

# CALIOP V4 Cloud Thermodynamic Phase Assignment and the Impact of Near-Nadir Viewing Angles

Melody A. Avery[1], Robert A. Ryan[2], Brian J. Getzewich[2], Mark A. Vaughan[1], David M. Winker[1], Yongxiang Hu[1], Anne Garnier[2], Jacques Pelon[3], Carolus A. Verhappen[2]

[1]NASA Langley Research Center, Atmospheric Composition Branch, Hampton, VA, 23681, U.S.
[2]Science Systems Applications Inc., 1 Enterprise Pkwy, Hampton, VA, 23666, U.S.
[3]Laboratoire Atmosphères, Milieux, Observations Spatiales, UPMC-UVSQ-CNRS, Paris, France.

*Correspondence to*: Melody A. Avery (melody.a.avery@nasa.gov)

**Abstract**

Accurate determination of thermodynamic cloud phase is critical for establishing the radiative impact of clouds on climate and weather. Depolarization of the Cloud-Aerosol Lidar with Orthogonal Polarization (CALIOP) 532 nm signal provides a useful addition to other methods of thermodynamic phase discrimination that rely on temperature, cloud top altitude or a temperature-based cloud phase climatology. Active detection of the thermodynamic phase of multiple cloud layers in a vertical column using cloud layer-integrated depolarization and backscatter also alleviates ambiguities in cloud phase determination by passive

radiometers. The CALIOP phase algorithm primarily uses vertically integrated cloud layer depolarization and attenuated backscatter to determine the dominant thermodynamic phase of hydrometeors present in a cloud layer segment, at horizontal resolutions for cloud layer detection varying between 333 m and 80 km, with cloud layer vertical resolutions between 60m – 8km. CALIOP ice cloud backscatter observations taken with a 0.3° near-nadir view between June 2006 and November 2007 include a significant amount of specular reflection from hexagonal smooth crystal faces that are oriented perpendicularly to the incident lidar

beam (HOI). These specular reflections from HOI are shown here to occur between 0 and -40 °C, with a peak in the CALIOP distribution observed globally at -15 °C. Recent viewing angle testing occurring during 2017 at 1°, 1.5° and 2° and reported here quantify the impact of changing the viewing angle on these specular reflections and verify earlier observations by POLDER. These viewing angle tests show that at the -15 °C peak of the HOI distribution the mean backscatter from all ice clouds decreases by 50% and depolarization increases by a factor of 5 as the viewing angle increases from 0.3° to 3°. To avoid these specular reflections,

the CALIOP viewing angle was changed from 0.3 to 3° in November 2007 and since then CALIOP has been observing clouds almost continuously for 12-13 more years. This has provided more data for a thorough re-evaluation of phase determination and has motivated changes to the CALIOP cloud phase algorithm for Version 4 (V4). The V4 algorithm now excludes over-identification of HOI at 3°, particularly in cold clouds. The V4 algorithm also considers cloud layer temperature at the 532 nm centroid and has been streamlined for more consistent identification of water and ice clouds. In V4 some cloud layer boundaries

have changed because 532 nm layer-integrated attenuated backscatter in V4 has increased due to improved calibration and extended layer boundaries, while the corresponding depolarization has stayed about the same. There are more V4 cloud layers detected, and combined with increasing cloud edges the V4 total atmospheric cloud volume increases by 6-9% over V3 for high-confidence cloud phases and by 1-2% for all cloudy bins. Collocated CALIPSO Imaging Infrared Radiometer (IIR) observations of ice and water cloud particle microphysical indices complement the CALIOP ice and water cloud phase determinations.



## 1. Introduction

Cloud ice crystals and water droplets have very different absorption and scattering properties (Sassen, 1991) and so accurate knowledge of the thermodynamic phase of clouds is needed for characterizing the transfer of radiation through Earth's cloudy atmosphere. Radiative transfer is in turn fundamental to passive remote sensing, and critical for assessment of cloud and aerosol

particle impacts to the Earth's climate (Cessana and Storelvmo, 2017). However, cloud particle formation and associated thermodynamic phase can be complicated by many factors, such as the interrelationship between ice particle nucleation and temperature, water vapor saturation, and ice nuclei availability in Earth's atmosphere (Lawson et al., 2010; Jensen et al. 2015; Kramer et al., 2016). While significant overlap between water and ice in the temperature range between 0 °C and -40 °C has been observed *in situ* (for example, Baker and Lawson, 2006), climate models often assume a climatological distribution of ice and

water in clouds that is based only on temperature (Jiang et al., 2012; McCoy et al., 2016 and references therein). As described in McCoy et al. (2016), varying the temperature of ice and water assignment can create substantial errors in radiative transfer calculations and significantly impact cloud climate feedback estimates and passive sensor optical thickness retrievals (Li et al., 2015, Yi et al., 2016).

Global remote sensing of clouds has proven to be valuable as a means for evaluating modelled cloud phase assignment and for developing ice and water cloud climatologies. NASA's A-Train constellation of polar-orbiting satellites has provided a complementary group of cloud-observing instruments, including the Moderate Resolution Imaging Spectroradiometer (MODIS) instruments on the Aqua and Terra satellites, the POLarization and Directionality of the Earth's Reflectances (POLDER) radiometer on PARASOL, the Cloud Profiling Radar (CPR) on CloudSat and the Cloud-Aerosol Lidar with Orthogonal

Polarization (CALIOP) and the Imaging Infrared Radiometer (IIR) on the Cloud Aerosol LIdar Pathfinder Satellite Observation (CALIPSO) satellite. While methods to infer thermodynamic phase have been developed for use with passive radiometers using infrared (IR) and shortwave measurements, the assessment of these methods relies upon comparisons with those from other passive sensors (Giraud et al. 2001; Riedi et al. 2010) or with CALIOP (Heidinger et al. 2010; Baum et al. 2012, Merchant et al., 2016). In recent years, the evaluation of methods for determination of thermodynamic cloud phase for satellite-based passive sensors has

relied increasingly on the cloud phase from CALIOP as the validation standard (Marchant, 2016; Baum et al., 2012; Yi et al., 2017).

Operational passive cloud retrieval methods generally assume that a single cloud layer exists in a field of regard, and this assumption is invalid for multi-layered cloud and aerosol scenes in which the uppermost ice cloud or aerosol layer(s) are optically

thin (Meyer et al. 2013; Wilcox et al. 2009; Wind et al., 2010). Mixed phase clouds containing supercooled water also produce ambiguities, and ice particles that form tend to grow very quickly and precipitate from the cloud. The inference of cloud phase from MODIS is also complicated over very cold and very bright surfaces, such as over ice and snow (King et al., 2004). Thermodynamic phase assignment in the current MODIS Collection 6 release of cloud optical properties is much improved (Platnick et al., 2017), however the daytime algorithm phase testing requires six independent retrievals for each pixel using

wavelength pairs in the shortwave infrared (SWIR) combined with the IR window channel. While microwave radars for large particle and precipitation detection preceded lidars in using polarization to deduce cloud microphysics (Schotland et al., 1971), radar retrievals of cloud microphysical properties for the relatively longer-wavelength CPR use a temperature-based climatology to approximate the ratio of ice vs water in observations made in bins between 0 °C and -40 °C (Wood, 2008).



Active and passive remote sensors that can detect linear polarization changes in scattered light from water and ice clouds have the advantage of determining thermodynamic phase without depending on temperature, pressure or detailed assumptions about ice particle habits. The POLDER passive radiometer measures cloud top reflectance from multiple visible and near-IR wavelengths at a variety of angles (Riedi et al., 2001, Riedi et al., 2010). The POLDER cloud phase index is determined from the angular

dependence of linearly polarized reflectance of 865 nm light from a single cloud top (Goloub et al., 2000). The angular signature in the polarized reflectance detects the difference in shape between spherical water droplets and non-spherical ice particles, regardless of particle size. However, the POLDER phase determination technique has difficulty with very thin cirrus clouds, cloud edges and especially with multiple cloud layers in a single scene (Goloub et al., 2000).

CALIOP's ability to directly measure vertically resolved changes in the linear polarization state caused by cloud particle scattering and internal refraction provides an advantage over other remote sensors that do not have this extra piece of information (Ceccaldi et al., 2013). Vertical profiling capability allows detection of multiple cloud and aerosol layers in a profile, until the lidar signal is completely extinguished. Ice and water clouds occurring in the same column can be differentiated using linear polarization changes and layer-integrated backscatter (Hu et al., 2009) to determine phase independently from the atmospheric state and particle size.

Further, high resolution range-resolved cloud profiles show vertical variations in cloud extinction coefficients that drive advanced studies of the impact of multiple atmospheric parameters on the development of clouds (Mace et al., 2007; Heidinger et al., 2015; Platnick et al., 2017; Subrahmanyam and Kumar, 2017; Iwabuchi et al., 2018).

This paper describes upgrades and changes to cloud phase and phase confidence assignments by the CALIOP, the first-ever space-

based polarization-sensitive lidar. CALIOP has been operating globally since June of 2006 (Winker et al., 2010) on the CALIPSO satellite. CALIOP transmits linearly-polarized light at 532 nm and 1064 nm. The total backscattered 1064 nm return is measured using a single avalanche photodiode. The 532 nm backscattered light passes through a polarization-sensitive beamsplitter, and then two photomultiplier tubes detect components that are polarized in a parallel and a perpendicular sense relative to the polarization plane of the transmitted laser light (Hunt et al., 2009). The CALIPSO satellite also carries the Imaging Infrared Radiometer (IIR),

which provides radiance measurements at three wavelengths in the IR window channel (8-12 μm) in a swath with the center collocated to the lidar ground track.

In this paper, we explain how CALIOP global cloud thermodynamic phase discrimination is accomplished (Sect. 2). We also discuss specular reflections from horizontally oriented ice (HOI) and show the impact to observations of backscatter, depolarization

and HOI as the CALIOP near-nadir viewing angle is varied between 0.3 ° and 3 ° to validate earlier POLDER observations. Due to CALIOP viewing angle and other changes, we show why Version 4 (V4) phase algorithm changes were needed. In Sect. 3, we describe the V4 phase algorithm in detail, and then we show the resulting V4 global distribution of water and ice clouds. We also show supporting observations of the water and ice microphysical indices observed by the collocated IIR. In Sect. 4, we show the impact of these algorithm changes, first by using a case study, and then by expanding this to investigate the global distribution of

water and ice clouds. A comparison of the cloud thermodynamic phases reported in V4 with those in Version 3 (V3) demonstrates the changes in ice and water cloud distribution that CALIOP data users can expect to find. In Sect. 5, we summarize these results.



## 2. CALIOP cloud phase determination

### 2.1 Overview of cloud phase determination by lidar

The first laboratory and atmospheric lidar observations of linearly polarized visible light scattering from hydrometeors were published by Schotland et al. in 1971. They observed that there was little change in the linear polarization state of laser light

backscattered at 180 ° from spherically symmetric water droplets, while the backscattered light from irregularly shaped ice crystals exhibited a significant amount of rotation in the plane perpendicular to the initial incident polarization. Specular reflections from smooth, hexagonal ice crystal faces that are oriented perpendicularly to a probing lidar beam do not depolarize the incident light and can create ambiguity in differentiating between ice and water clouds using depolarization alone (Platt, 1977). As mentioned by Platt (1977), solving this ambiguity requires using observations of backscatter in addition to depolarization to diagnose

thermodynamic cloud phase. These two fundamental observations drive the lidar cloud particle phase discrimination algorithm that is employed by CALIOP to separate global cloud observations into ice and water clouds.

Adaptation of ground-based and airborne polarization lidar techniques to a space-based platform necessitated careful consideration of multiple scattering effects. Hu, 2007a and Hu et al., 2007b used a Monte Carlo method to model the impact of multiple scattering

on the polarization state of the return signal from clouds containing water droplets for a spaceborne lidar such as CALIOP. These studies found that multiple scattering in a water cloud causes depolarization of a linearly polarized lidar beam that increases with the cloud layer optical thickness. In contrast, internal refraction and reflection in randomly oriented ice (ROI) crystals create a depolarization signal that is mainly independent of cloud optical thickness (Hu et al., 2001). Additionally, specular reflections from horizontally oriented hexagonal plates of ice crystals, hereafter called "horizontally oriented ice" (HOI) create backscatter that can

be quite large without causing significant depolarization of the lidar signal. Using simulations with this Monte Carlo model and CALIOP observations, Hu et al. 2007a and Hu et al., 2009 demonstrated that the distribution of cloud layer-integrated 532 nm backscatter coefficients and layer-integrated volume depolarization ratios provides a relatively simple and effective threshold method for separating clouds into three categories: ROI, HOI and water. The following section provides an overview of how the cloud thermodynamic phase algorithm fits into the CALIOP data processing stream and describes how these quantities are derived

from the 532 nm parallel and perpendicular backscatter signals detected by CALIOP.

### 2.2 Cloud phase determination by CALIOP

In the CALIOP Level 1 data processing stream, the lidar 532 nm parallel and perpendicular polarization signal profiles are first calibrated to provide attenuated backscatter in each channel (Powell et al., 2009, Getzewich et al., 2018, Kar et al., 2018). Then as

a first step in Level 2 processing, a feature detection algorithm isolates locations of elevated backscatter in the signal return profiles from 30 km to the surface, at horizontal along-track averaging resolutions including 333m single laser shots from the surface to 8.2 km, and otherwise ranging from 1 to 80 km (Vaughan et al., 2009). The topmost "features" are classified as clouds or aerosols based on a cloud-aerosol discrimination algorithm (CAD) that is described in detail for V4 in Liu et al., 2019. The cloud phase algorithm is then applied to features that have been identified by the CAD as clouds. Cloud layers detected at 333 m and 1 km are

assigned a thermodynamic phase, but additional optical properties are not retrieved for these layers. For layers detected by horizontal averaging of 5 km or more, extinction coefficients are retrieved as described in Young et al. (2018). For transparent layers, the attenuated backscatter coefficients in the profile below the layer base are renormalized to account for the consequent overhead signal attenuation. After this backscatter renormalization, the algorithms work downwards, (re)classifying lower-level





features and repeating the extinction retrieval renormalization process, until the lidar signal is completely attenuated, or the surface is encountered (Young and Vaughan, 2009).

For all of the cloud layers detected, a layer-integrated attenuated 532 nm backscatter ($\gamma'_{532}$) is calculated by integrating the total 532 nm attenuated backscatter coefficients $\beta'_{532} = \beta'_{532,\perp} + \beta'_{532,\parallel}$ through the depth of the layer, where the subscripts $\perp$ and $\parallel$ represent, respectively, measurements made in the perpendicular and parallel channels. $\gamma'_{532}$ is estimated using Eq. (1), with z representing the lidar range bins:

$$\gamma'_{532} = \frac{1}{2}\left(\sum_{k=top+1}^{base}(z_{k-1} - z_k)\left(\beta'_{532}(z_{k-1}) + \beta'_{532}(z_k)\right)\right) - \frac{1}{2}\left((z_{top} - z_{base})\left(\beta'_{532}(z_{top}) + \beta'_{532}(z_{base})\right)\right) \ (1)$$

Ideally the depolarization ratios used in cloud phase determination would be determined from only particulate backscatter ($\delta_p$).
However, since the cloud layer optical properties needed to separate molecular from particulate depolarization in the elastic backscatter signal have not yet been calculated at this stage of the CALIOP data processing, $\delta_p$ is not available. Instead the phase algorithm assumes that the 532 nm molecular backscatter contribution is small compared to the particulate backscatter for all layers with $\gamma' > 0.01$ sr$^{-1}$, so that the molecular contribution to the parallel channel backscatter can be neglected, and $\delta_p(z) \cong \delta_v(z)$, with $\delta_v$ the range-resolved volume depolarization ratio (Hu et al., 2009). The layer-integrated volume depolarization ratio is
defined as the ratio of the mean perpendicular and parallel signals through the cloud layer as shown in Eq. (2):

$$\delta_v = \frac{\langle \beta'_{532,\perp}(z)\rangle}{\langle \beta'_{532,\parallel}(z)\rangle} \qquad (2)$$

To illustrate the clustering of cloud layers by thermodynamic phase in $\delta_v$ vs $\gamma'_{532}$ space, Figure 1 shows the distribution of 5 km segments of V4 cloud layers identified by the CAD and assigned a thermodynamic phase by the phase algorithm at a 0.3°
viewing angle in 2007, and at 3° in 2008. Between 6-12% of layers identified by the CAD as cloud layers are assigned as phase "unknown" and are not shown in Fig. 1.

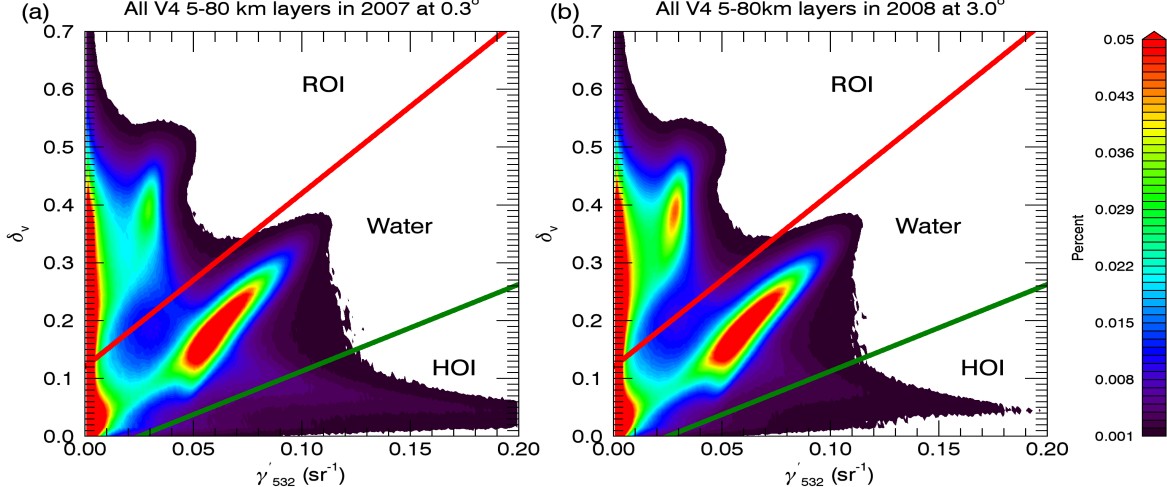

**Figure 1:** Panels (a) and (b) show the layer-integrated volume depolarization ratio ($\delta_v$) vs 532 nm integrated attenuated backscatter ($\gamma'_{532}$) for
all 5 km cloud layer segments detected at 5 to 80 km averaging length, globally from 0 to 20 km, daytime and nighttime from the V4 standard 5 km layer product. Panel (a) shows the distribution of 36 million 5 km cloud layer segments observed at 0.3° near-nadir viewing angle in 2007, which includes the majority of the year until late November, except for periods of viewing angle testing.  Panel (b) shows 49 million 5 km cloud segments detected at 3° during 2008. The red line represents the threshold between the randomly oriented ice and water sectors, and the green





line shows the water-horizontally-oriented ice threshold. Layers with hexagonal plates can occasionally have $\gamma'_{532}$ up to 1.0 sr[-1], which is off the scale shown here.

Figure 1 shows the thresholds for ice vs water as solid red and green lines. As determined from the Hu Monte Carlo calculations (Hu et al., 2007(a)), the equation for the ROI-water threshold (red line in Fig. 1) is

$$\delta_v = 3.0\gamma'_{532} + 0.12 \quad (3)$$

The HOI-water threshold (green line in Fig. 1) is

$$\delta_v = 1.5\gamma'_{532} - 0.0375 \quad (4)$$

Because the $\delta_v$ approximation for $\delta_p$ becomes less accurate for optically thin cloud layers, the phase algorithm estimates particulate depolarization for layers with $\gamma'_{532} < 0.01$ sr[-1]. The estimation assumes that the particulate backscatter from ice cloud particles at the 532 nm and 1064 nm channels are equivalent, with a backscatter color ratio = 1 (Vaughan et al., 2010), and that molecular backscatter is negligible at 1064 nm (Hu et al., 2009). It is further assumed that molecular depolarization is small (Hunt et al., 2009), so that the molecular contribution to 532 nm perpendicular channel backscatter is negligible. This approximation is expressed by Eq. (3) from Hu et al., 2009 as

$$\delta_p \approx \frac{\beta_{532,\perp}}{\beta_{1064,\parallel}} \approx \frac{\beta_{532,\perp}}{\beta_{1064} - \beta_{532,\perp}} \approx \frac{1}{\frac{\beta_{1064}}{\beta_{532,\perp}} - 1} = \delta_{1064} \quad (5)$$

The layer-integrated depolarization used in the phase algorithm is approximated particulate depolarization, either $\delta_v$ from Eq. (2) or $\delta_{1064}$ from Eq. (5). To avoid confusion with $\delta_p$ reported in the standard Level 2 layer products, hereafter we will use the notation $\delta_{p,eff}$. This approximation causes a subtle shift in the distribution for layers with $\gamma'_{532} < 0.01$ sr[-1] relative to $\delta_v$. One can often observe this as a discontinuity in the figures that follow at $\gamma'_{532} = 0.01$ sr[-1]. The phase algorithm assigns a confidence level of "high", "medium", "low" or "none" to the thermodynamic phase assignment of a cloud layer. For clouds to be classified as ROI, HOI or water with "high confidence", they must fall into one of the three sectors divided by these thresholds in $\delta_{p,eff}$ vs $\gamma'_{532}$ space. These figures will be referred to in this paper as "Hu phase diagrams".

**2.3 CALIOP detection of HOI**

CALIOP detection of specular reflections from HOI presents a special case because the frequency of these reflections varies depending on the CALIOP near-nadir viewing angle. In earlier versions of the CALIOP phase detection algorithm, this viewing angle dependency was not included. In this section, we first describe some background information about specular reflections from HOI, and then how CALIOP observations of HOI change as the viewing angle is changed. The following section (Sect. 2.4) describes the consequent change to the V4 phase algorithm.

Global POLDER observations of polarized reflectance from cloud tops at multiple viewing angles from the ADEOS satellite show that the majority of specular reflections from HOI occur from hexagonal plates that are oriented horizontally with a tilt angle of less than 1° with respect to a nadir view (Chepfer, 1999, Breon and Dubrelle, 2004). These hexagonal plates typically occur between -10 °C and -35 °C as noted by Noel and Chepfer (2010), and as predicted by the ice crystal habit classification of Bailey and Hallet (2009). Relatively large hexagonal plates tend to align nearly horizontally under conditions of low updraft velocity and typical atmospheric turbulence (Platt, 1978). When growing in a supersaturated environment with low updraft velocities, these ice crystals tend to be more pristine than those particles that grow in convective environments, and thus have smoother faces. Occasional observations of specular reflections from hexagonal plates or hexagonal faces of aggregates lofted by convection have been made at colder temperatures during aircraft field campaigns (Sassen and Takano, 2000, Mioche et al., 2010) and unpublished





observations from SEAC4RS in 2013). However, these appear infrequently in the global data. As reported by Breon and Dubrelle (2004), the POLDER observations suggest that increasing the nadir viewing angle beyond 1° will gradually reduce observations of HOI until they become rare at a 3° viewing angle.

At the start of the CALIPSO mission, CALIOP's viewing angle was set to 0.3° off-nadir, which minimized specular reflections from still waters while retaining the ability to readily detect HOI (Hunt et al., 2009). After five months of obtaining CALIOP 0.3° off-nadir data, testing of a CALIOP nadir viewing angle of 3° was initiated with the objective of minimizing CALIOP detection of specular reflections from HOI. The engineering trade was between detection of HOI or more accurate discrimination between ice and water. A further motivation was to eliminate potential difficulties with extinction retrievals for cloud layers containing HOI

(Hu, 2009). At the end of November 2007, after a first set of viewing angle tests, the CALIOP viewing angle was changed to 3°, greatly reducing the amount of HOI evident in the global data set as predicted. After nearly ten years of science data collection at a nominal viewing angle of 3°, the CALIPSO project performed a series of additional CALIOP measurements at 1.0°, 1.5° and 2.0° nadir viewing angles during 2017. The motivation for this testing was to inform the choice of an off-nadir viewing angle for future space lidar missions, given the engineering trade between avoiding specular reflections in the ice cloud return signals and

avoidance of oblique surface return signals.

Table 1 shows a comprehensive history of CALIPSO viewing angles maintained for more than 24 hours during the first 12 years of the CALIPSO mission. For reference the table shows the dates of a first set of tests in 2006-2007 in the first three rows, and more recent viewing angle test dates in the second set of three rows.

| Viewing Angle | 0.3° | 3° | 0.3° | 3° | 0.3° | 3° |
|---|---|---|---|---|---|---|
| Start Date | 06/13/2006 | 11/07/2006 | 11/15/2006 | 08/21/2007 | 09/07/2007 | 11/28/2007 |
| End Date | 11/07/2006 | 11/15/2006 | 08/21/2007 | 09/07/2007 | 11/28/2007 | 02/02/2017 |
| Viewing Angle | 2.0° | 3° | 1.0° | 3° | 1.5° | 3° |
| Start Date | 02/02/2017 | 02/08/2017 | 02/22/2017 | 02/28/2017 | 06/01/2017 | 06/26/2017 |
| End Date | 02/08/2017 | 02/22/2017 | 02/28/2017 | 06/01/2017 | 06/26/2017 | ongoing |

**Table 1:** CALIPSO Off-nadir viewing angle history listing the dates during which the CALIPSO viewing angle was maintained for more than 24 hours. In addition to these extended near-nadir viewing angle changes, a series of rapid tests were performed at more oblique viewing angles of 10, 14 and 30 degrees during 2014-2017. This testing is not reported here, and is omitted from the analysed data, as are any other short periods of time when the CALIPSO satellite or CALIOP were performing brief special operations. CALIPSO data users can determine the off-nadir

viewing angle used for data collection by examining the parameter "Off_Nadir_Angle" in CALIOP Level 1 Profile or Level 2 Layer files.

A challenge in analyzing the impact of viewing angle changes on thermodynamic phase assignment is that these maneuvers can only be performed in series. So, natural variability in the ensemble of clouds sampled during observations at each viewing angle cannot be controlled in the analysis. Nonetheless, the viewing angle testing provides useful information, and the results are described here since they provide context for changes to the new V4 phase algorithm. Given the sharp peak in reflectance at visible

wavelengths near 0° as measured by POLDER (Fig. 3 of Breon and Dubrulle, 2004), a hypothesis was that the specular reflections would drop off sharply as the viewing angle was increased. Figure 2 shows that the fraction of observed high-confidence HOI drops as the CALIOP viewing angle increases, just as expected from the POLDER observations and by earlier airborne lidar observations (Sassen and Noel, 2005).

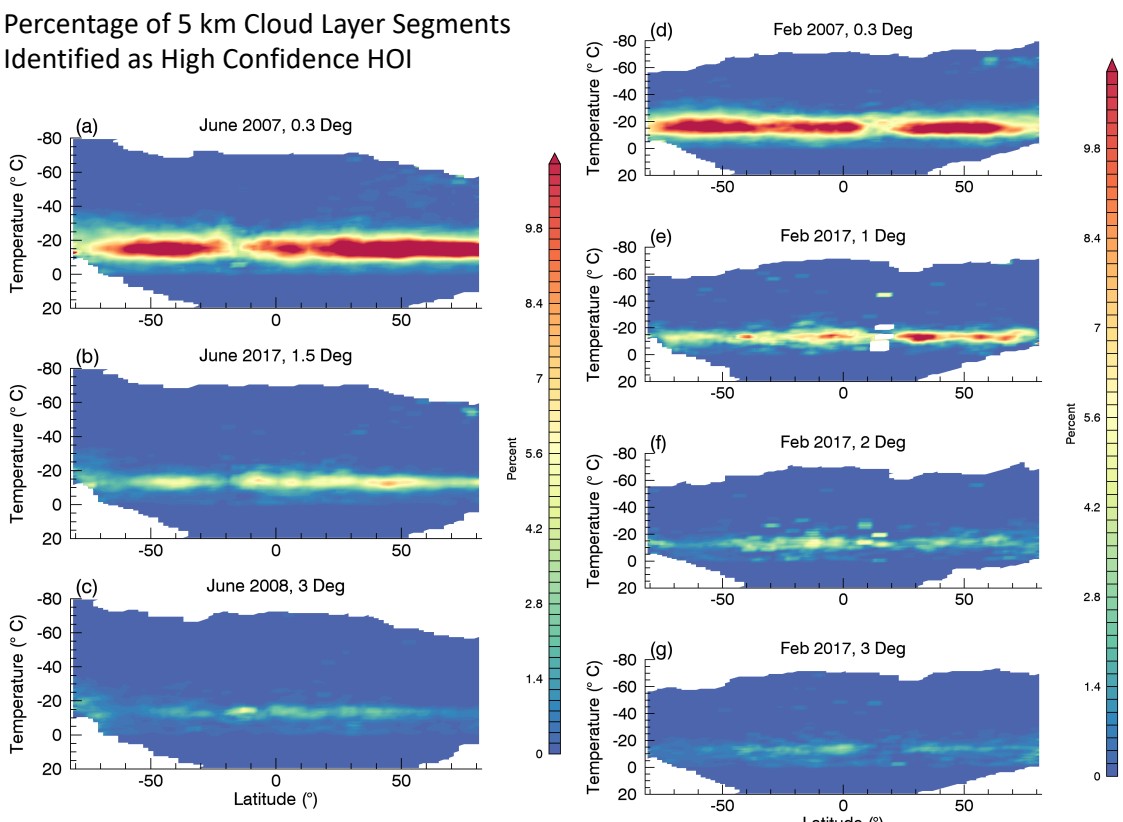

**Figure 2:** All panels show the distribution of the percentage of all clouds that are detected by CALIOP in V4 as high-confidence HOI, as a function of latitude and the MERRA-2 re-analysis temperature. This analysis uses all V4 5 km cloud layer segments detected globally, daytime and nighttime, with averaging length of 5-80 km and 532 nm backscatter centroid altitudes between 0 and 14.5 km. Temperatures are from the MERRA-2 re-analysis and are interpolated to the height of the 532 nm backscatter centroid for a cloud layer segment, see Eq. (6) for details.

Since the testing at viewing angles of 1°, 1.5° and 2° was done in both February and June of 2017, reference plots at 0.3° and 3° are shown for both months to minimize mixing seasonal differences into the viewing angle comparisons. The left-hand column of Fig. 2 shows observations made in June. Observations made at 1.5° in June 2017 took up almost the entire month, and so the comparison plots at 0.3° and 3° are shown for 2007 and 2008, respectively. The righthand column of Fig. 2 shows tests at 1° and 2° from February 2017, as well as a comparison with three observations taken during the rest of that month. February 2007 is shown for a 0.3° reference. Although there is some scatter in the HOI detected, these plots show a steady reduction of HOI observations as the viewing angle increases, validating that the space-based CALIOP reproduces earlier results from POLDER. There is an order of magnitude reduction in HOI observed by CALIOP between viewing angles of 0.3° and 3°, information that is used to adjust the V4 CALIOP phase algorithm.

The primary impacts of specular reflections on CALIOP data are elevated backscatter and reduced depolarization. Figure 3 illustrates these changes as they are influenced by temperature and by viewing angle. The cloud layer temperatures shown in Fig. 3 are cloud layer centroid temperatures (Garnier et al., 2015). In V4 these are MERRA-2 re-analysis temperatures interpolated to the 532 nm attenuated backscatter centroid, defined as:





$$z_{cent} = \frac{\sum_{j=top}^{base} z_j \beta'_{532}(z_j)}{\sum_{j=top}^{base} \beta'_{532}(z_j)} \quad (6)$$

Figure 3(a) shows the percent of all clouds identified as HOI for a cloud centroid temperature range of 0 °C to – 40 °C. The mean attenuated backscatter (Fig. 3(b)) and volume depolarization (Fig. 3(c)) for ice cloud layers with $\gamma'_{532} \geq 0.02$ sr$^{-1}$ are then shown from the corresponding time periods.

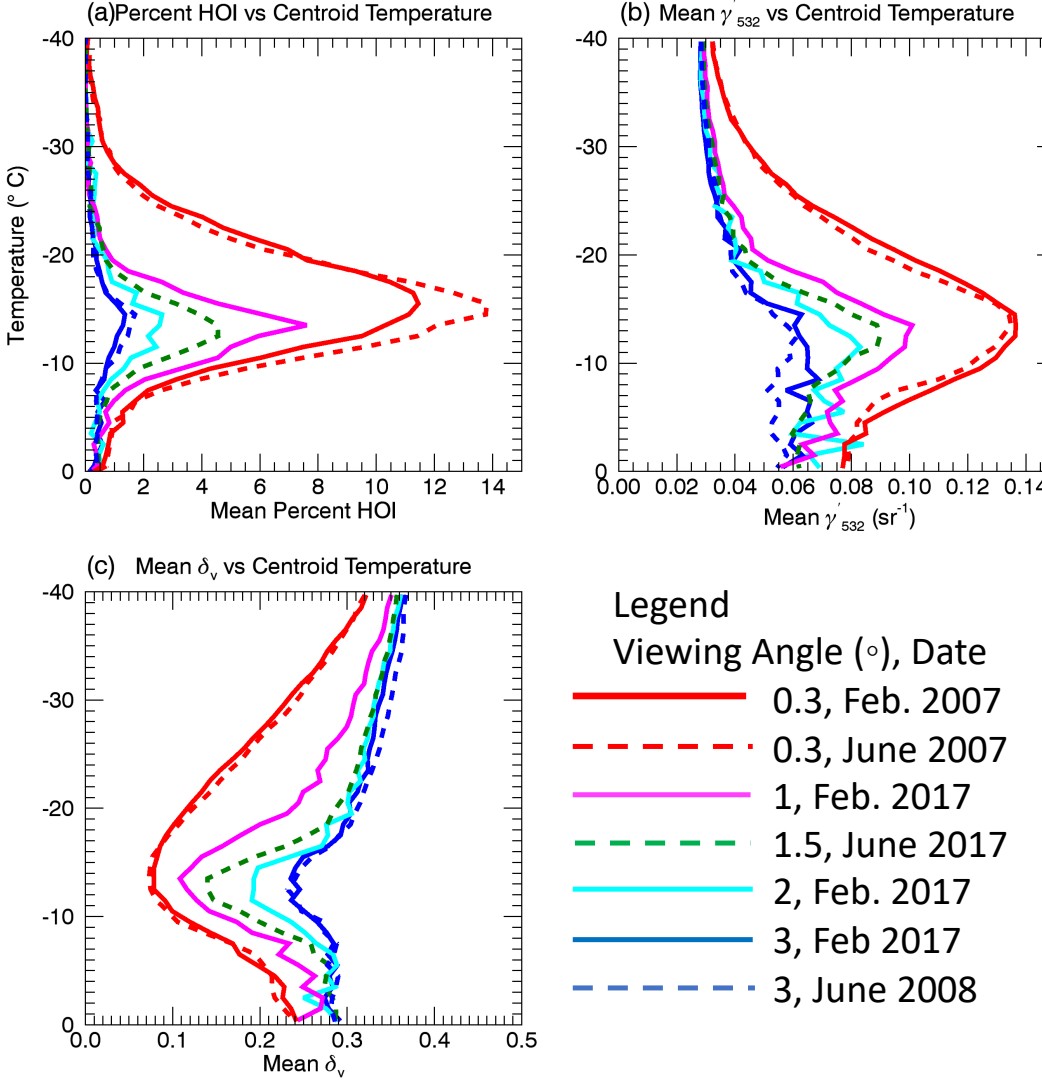

**Figure 3:** Panel (a) shows the global mean percentage of all clouds detected as HOI in V4 by CALIOP as a function of the 532 nm cloud layer centroid temperature. The V4 mean $\gamma'_{532}$ for ice clouds (ROI+HOI) with $\gamma'_{532} > 0.02$ sr$^{-1}$ is shown in Panel (b) and the V4 mean $\delta_v$ in Panel (c).

The plots in Fig. 3 show that the peak of the distribution of HOI observed by CALIOP occurs at centroid temperatures of about

10    -12 to -15 °C, and that the impact of specular reflections from HOI on $\gamma'_{532}$ and $\delta_v$ measurements of ice clouds decreases steadily with increased nadir viewing angle.





**2.4 Changes to the CALIOP phase algorithm between V3 and V4**

Given the order of magnitude reduction in the detection of specular reflections from HOI at 3°, it is reasonable to expect that the phase algorithm might need some adjustment. The V3 CALIOP phase algorithm was designed to accommodate the detection of HOI and mixed ROI/HOI cloud layers by implementation of a spatial coherence test described in detail in Hu et al., 2009. This test

is based on the observation that $\gamma'_{532}$ and $\delta_{p,eff}$ are positively correlated in water clouds, but negatively correlated in HOI clouds. The spatial coherence test uses the correlation observed between $\gamma'_{532}$ and $\delta_{p,eff}$ in horizontally-adjacent cloud layers to sort water layers from HOI. While the test is useful at 0.3°, ten years of data acquired at 3° showed that the spatial coherence test was identifying HOI layers too aggressively. Originally this test was designed to correct for ambiguity in locating layers containing HOI that appear in the water sector. However, V3 of the phase algorithm implements this coherence test for three sequential cloud

layers detected with a horizontal averaging length of 5 km or less at both 0.3° and 3° in all three phase sectors.

HOI detected by the phase algorithm in V3 is shown in Figure 4. The plots show layers identified as HOI by the spatial coherence test in the ROI or water sectors as well as high-confidence HOI in the HOI sector. Panel (a) shows HOI layers identified in 0.3° data from 2007 and Panel (b) shows 3° data from 2008.

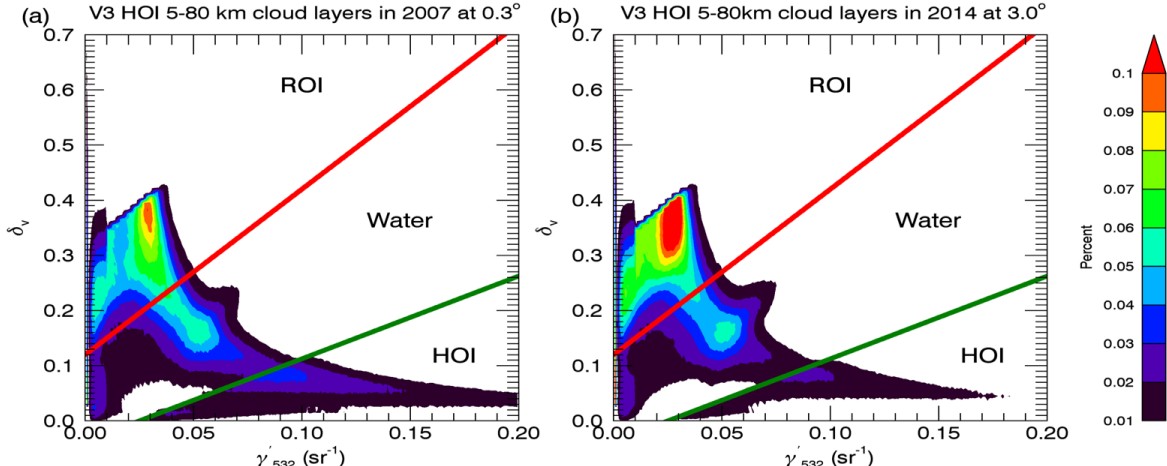

**Figure 4:** Comparison of V3 HOI for both the nadir (0.3°) and "tilted" (3°) viewing angles. HOI identified by the spatial coherence test and high confidence HOI distributions are plotted together on basic Hu phase diagrams of $\delta_v$ vs. $\gamma'_{532}$. The red lines designate the threshold between ROI and water, and the green lines show the threshold between water and HOI.

The reduction in high-confidence HOI from the 0.3° cloud observations in 2007 to the 3° observations in 2008 is shown by the reduced population below the green line in Fig 4(b) when compared with Fig. 4(a). As mentioned above, Hu et al. (2009) describes an ROI/HOI mixing line, extending from a maximum in the ROI sector (maximum in the distribution above the red line) through the water sector to the HOI high $\gamma'_{532}$, low $\delta_{p,eff}$ "tail", a negative $\gamma'_{532}$ and $\delta_{p,eff}$ relationship. A positive correlation between $\gamma'_{532}$ and $\delta_{p,eff}$ causes a "lump" in the distribution in the water sector. The spatial continuity test was designed to differentiate between

water and mixed ROI/HOI only in the water sector. However, in V3 the test is applied to layers in all three sectors and identifies a significant population of HOI at a 3° view in the ROI sector that dominates the distribution of apparent HOI. Further, when the scale on Fig. 4(b) is adjusted to emphasize the HOI identified by spatial continuity, the positive correlation between backscatter and depolarization can be seen to be dominant in the water sector, indicating a significant misidentification of water as HOI.



Since it is clear from the viewing angle tests shown above that the 3° view greatly reduces the amount of specular reflections from horizontally-oriented planar ice crystal faces, it is likely that the V3 algorithm is mistaking ROI for HOI in the ROI sector as well as water for HOI in the water sector because of natural variability or variation due to noise occurring in $\gamma'_{532}$ and $\delta_{p,eff}$. Based on these observations, we decided to perform the spatial coherence test in V4 only on data taken with viewing angles of < 1°, layers

with $\gamma'_{532}$ > 0.02 sr$^{-1}$ and in the water sector. Consequently, the amount of HOI identified in V4 in 3° data is much reduced from V3.

The phase algorithm prior to V4 also performed a series of secondary tests based on cloud top temperatures. This was done to ensure that clouds with tops > 0 °C were correctly identified as water clouds, and clouds with tops < -40 °C were identified as ice

clouds (Hu et al., 2009). In V4, the temperature testing is improved by using the temperature interpolated to the 532 nm attenuated backscatter centroid instead of the cloud top temperature, as defined by Eq. (6) in the previous section. In V4 the MERRA-2 reanalysis temperature field is used consistently throughout the 13-year data set. In contrast, for meteorological data V3 uses output from a series of GEOS-5 model versions with physics that were updated by the GMAO over time. Use of the MERRA-2 temperature field interpolated to the location of the 532 nm backscatter centroid has the dual advantages of using a temperature

field produced by consistent reanalysis model physics throughout the CALIOP data record and phase decisions based on atmospheric conditions at the location where the majority of observed cloud particles are located. This algorithm change was necessary because the majority of backscatter detected from a mixed-phase cloud layer does not always come from the cloud top. For example, it is possible to have geometrically thick clouds with very cold cloud tops, but much lower and warmer apparent cloud bases. In these cases, the V3 phase algorithm could assign a thermodynamic phase based on more tenuous scattering from a thinner part of a cloud at the cloud top, rather than responding to more robust scattering closer to the cloud base. In V4, a cloud

with an optical centroid near the warm cloud base can be assigned as a water cloud, even if it has a very cold cloud top. An example of this is shown in Sect. 4.

### 3 The V4 cloud phase algorithm

### 3.1 Phase algorithm details

The flowchart in Figure 5 shows the decision tree used by the V4 cloud phase algorithm. The algorithm will assign a thermodynamic cloud phase, ice or water, with additional differentiation between ROI and HOI. If a cloud layer does not provide enough signal for unambiguous phase assignment, or there are other errors in layer detection, the phase algorithm will assign a cloud phase of "unknown". Additionally, the phase algorithm assigns a rating of high, middle, low, or none to characterize "confidence" in the phase assignment. As discussed in Sect. 2, the $\delta_{p,eff}$ - $\gamma'_{532}$ relationship is primary for determining high-

confidence phase assignments. Medium-confidence phase assignments are made for layers that are reassigned to a phase outside of their $\delta_{p,eff}$- $\gamma'_{532}$ sector, based on 532 nm centroid temperature (Eq. (6)), layer-integrated attenuated backscatter color ratio ($\chi' = \frac{\langle\beta'_{1064}\rangle}{\langle\beta'_{532}\rangle}$) or spatial coherence if detected at 0.3° with $\gamma'_{532}$ > 0.02 sr$^{-1}$. Occasionally, optically thin water layers with $\gamma'_{532}$ < 0.01 sr$^{-1}$ are assigned with a low-confidence rating. Details of the V4 phase and phase confidence assignments are described in this section.


Initially the cloud phase algorithm determines whether a feature has been identified sufficiently well as a cloud layer by the CAD algorithm. As described in Liu et al., 2019, CAD scores range from -100 to 100 with positive numbers designating clouds and





negative numbers designating aerosols. A feature with a CAD score of 100 is a cloud detected with the highest level of confidence, while CAD scores < 20 designate clouds detected with "no" confidence. Layers detected at a horizontal averaging length of 5 km or longer with a CAD score of < 20 are assigned with an "unknown" cloud phase and a corresponding phase confidence level of "none". These layers are often either optically thin, or occur underneath optically thick clouds, where the SNR is low due to signal

attenuation by passing through overhead features with a large cumulative optical thickness. CAD scores with absolute values > 100 indicate a special consideration. In the V4 phase algorithm a cloud layer with a CAD score of 103, signifying a layer with suspiciously large $\gamma'_{532}$, is also assigned as an "unknown" phase with no phase confidence. An exception to this occurs when a cloud layer has been detected by a single laser shot or at 1 km of horizontal averaging, and is assumed to be well-classified as a cloud.

Cloud layers that pass the initial CAD confidence test are divided into three groups; ROI, water and HOI based on their location relative to the thresholds in the $\gamma'_{532}$–$\delta_{p,eff}$ Hu phase space, as described in Section 2. The only remaining test for cloud layers with $\gamma'_{532}$ and $\delta_{p,eff}$ that fall into the ROI sector is to check that they have a centroid temperature that is colder than 0 °C. If so, they will be assigned as high-confidence ROI. Globally, between 40-50 % of cloud layers detected by CALIOP at 5-80 km are classified

this way. If the 532 nm centroid temperature is warmer than 0 °C, the cloud layer is re-classified as a water cloud, but with only medium confidence because for a small $\gamma'_{532}$ the depolarization is relatively high. Less than 1% of 5-80 km cloud layers must be re-classified using temperature, evidence that the Hu threshold method produces results that are in good agreement with basic thermodynamics.

When a cloud layer falls into the "HOI sector", the cloud layer $\gamma'_{532}$ is large, but $\delta_{p,eff}$ is very small. If the corresponding $\delta_{p,eff}$ is negative, in V4 the layer will be assigned as an unknown phase with no phase confidence level. Due to a coding error in V4.1 and V4.2 $\delta_{1064}$ was used for this test instead of $\delta_{p,eff}$, however this has been corrected in V4.5 and impacts only a small number of HOI sector layers with the 3° viewing angle. If the centroid temperature is > 0 °C the layer will be assigned as low-confidence water, which occurs only rarely, in < 0.1% of layers detected at 5-80 km. Otherwise cloud layers in the HOI sector will be assigned as

high-confidence HOI, regardless of the CALIOP viewing angle.





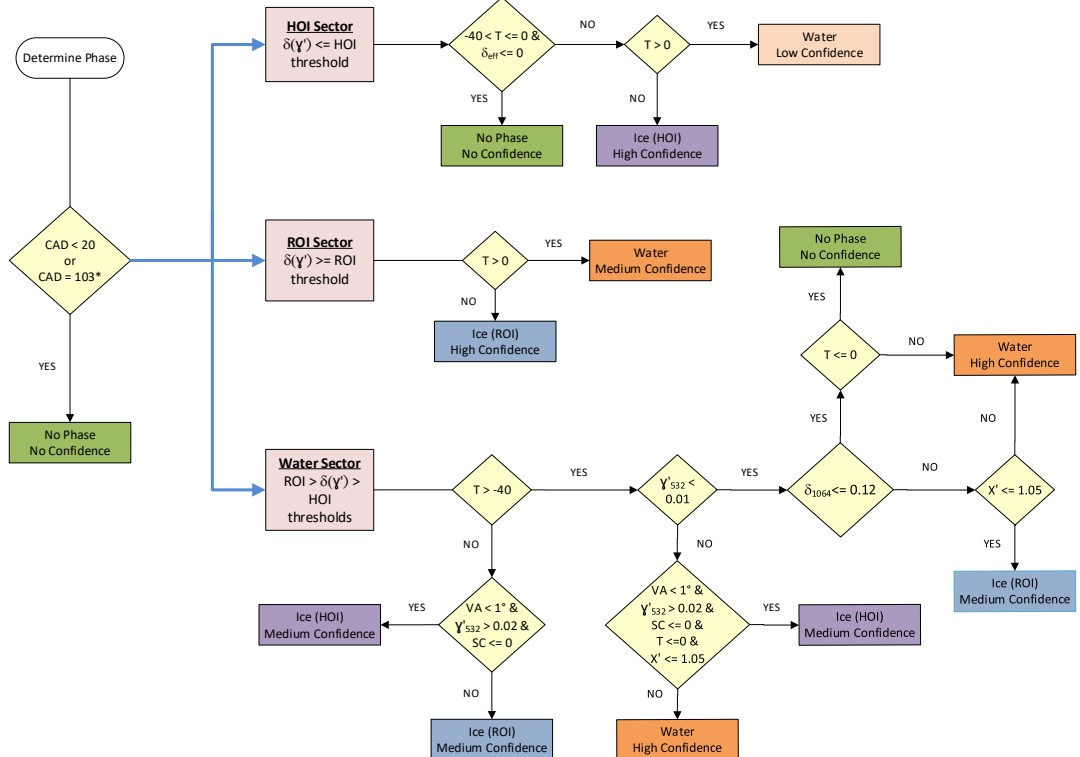

**Figure 5:** V4 phase algorithm decision tree. The first set of decisions after the initial CAD tests are based on the thresholds in the Hu $\gamma'_{532}$-$\delta_{p,eff}$ phase diagam described in Section 2. In this flow chart "T" refers to the 532 nm centroid temperature of a cloud layer in $°$C, $\gamma'_{532}$ is the 532 nm integrated attenuated backscatter, $\delta_{eff}$ is the estimated particulate depolarization using either the 1064 channel or $\delta_v$ as described in Section 2, VA is the near-nadir viewing angle, SC is the spatial coherence test result and $\chi'$ is the layer-integrated attenuated 1064/532 nm backscatter coefficient ratio.

The final and most complicated decision tree in the phase algorithm occurs for cloud layers in the water sector. This includes liquid water layers that are optically thick with enhanced depolarization due to multiple scattering. The phase algorithm first checks to see if these layers are colder than -40 °C. If they are, they are assigned as medium-confidence ROI. If the viewing angle is 0.3°, a

10 spatial coherence test described in Hu et al., 2009, is applied to identify medium-confidence HOI. If a water-sector cloud layer has a centroid temperature that is warmer than -40 °C, and it was measured at 3° with $\gamma'_{532} > 0.01$ sr$^{-1}$ the layer will be classified as high-confidence water. If the layer was measured with a viewing angle of 0.3°, has $\gamma'_{532} > 0.02$ sr$^{-1}$ and was detected at 5 km or less, there is a 5 % chance that it will also have negative spatial coherence. If such a layer also has a centroid temperature cooler than 0 °C and $\chi' < 1.05$ then it will be identified as medium-confidence HOI.

The water sector includes optically thin cloud layers that pass the CAD test with a CAD score > 20, but with $\gamma'_{532} < 0.01$ sr$^{-1}$ and a relatively low SNR, either because of large overhead optical depth or because they are tenuous. For these layers, $\delta_{p,eff}$=$\delta_{1064}$, and those with $\delta_{p,eff}$ greater or equal to 0.12 and $\chi' < 1.05$ are considered to be ice, as shown in Hu et al. (2009), Fig. 11. These layers are classified as medium-confidence ROI and occur as less than 0.25% of CALIOP layers detected at 5-80 km. If $\chi'$ is higher than





this threshold the layers are classified as high-confidence water. If $\delta_{p,eff}$ is less than 0.12 and the centroid temperature is warmer than 0 °C they are also high-confidence water, but if colder than 0 °C the phase assignment is "phase unknown with no confidence".

An additional adjustment to cloud/aerosol partitioning has been added in V4, to identify features that have been classified by the CAD algorithm as aerosols, but which occur in spatial proximity to ice cloud layers (Liu et al., 2019). This spatial proximity test is applied only to layers with base altitudes above 4 km.  The intention is to distinguish between optically thin, relatively weakly depolarizing cirrus layers and dust layers with particles that have a crystalline structure that depolarizes the lidar signal. These layers are designated as cloud "fringes" since they are identified at the margins of more robust clouds. Since they are classified outside of the CAD and cloud phase algorithms, they are identified as ROI but assigned a confidence level of "none". These "cloud

fringes" make up about 1.5% of cloud layers detected at 5-80 km, and since they tend to be optically thin, they add a negligible amount to the total detected cloud optical thickness. All ROI layers assigned as cirrus fringes outside of the CAD and phase algorithms are designated with a special CAD score of 106 (Liu et al., 2019).

### 3.2 Global maps of V4 high-confidence cloud phase occurrence

The CALIOP phase algorithm works efficiently to separate ice from water clouds. For example, in 2007 V4 data acquired with a

0.3° view, 73 % of 5 km cloud layer segments were identified as high-confidence ice or water, with 10 % HOI, as determined using the V4 standard 5km cloud layer product. In 2008, with a 3° view, the fraction of high-confidence cloud layers increased to 86-87 %, with negligible (< 1 %) HOI. Shown in Figure 4 are two global maps of the distribution of 5 km high-confidence ice and water cloudy bins detected during April 2008.

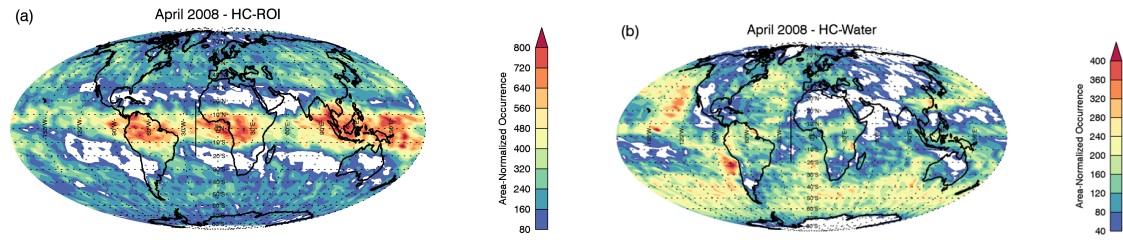

**Figure 6:** Maps of the area-normalized occurrence of V4 cloudy bins in 2° x 5° grid boxes from the surface to 20 km during April, 2008. Both daytime and nighttime standard V4 5 km cloud profiles were used to compile the occurrence of 5 km x 60 m bins from each cloud phase. Occurrence in each 2° x 5°grid box has been weighted to match a corresponding grid box at the equator to avoid oversampling at high latitudes. Panel (a) shows the global occurrence of high-confidence ROI.  Panel (b) shows the corresponding occurrence of high-confidence water. At altitudes below 8.2 km a 60 m bin is assigned as cloudy if either top or bottom 30 m segment contains a cloud.

Figure 6 shows two global maps of the distribution of 5 km high-confidence ice and water cloudy bins during April 2008 from 5km cloud profiles. Figure 6(a) shows the location of high-confidence ROI. ROI is the thermodynamic phase observed most often by CALIOP. ROI clouds occur throughout the global atmosphere, but are most prevalent in the tropics, and then again in the mid to high latitudes, with a pronounced minimum in the subtropics. Fig. 6(b) shows high-confidence water. Water clouds observed by

CALIOP occur most frequently at northern hemisphere high latitudes and over the southern hemispheric oceans. There are pronounced maximums in the high-confidence water cloud distribution observed off the west coast of continents at mid-latitudes where clouds are forming over cold ocean currents.

The change to a more oblique viewing angle of 3° off-nadir causes the CALIOP phase algorithm to identify a much-reduced population of HOI. Figure 7 uses global maps of the observed high-confidence HOI volume fraction of cloudy bins to show the striking reduction in specular reflections and HOI identification. Note that the scale on Figure 7 Panel (b) at 3° is ten times smaller

5   than that of Panel (a) at 0.3°.

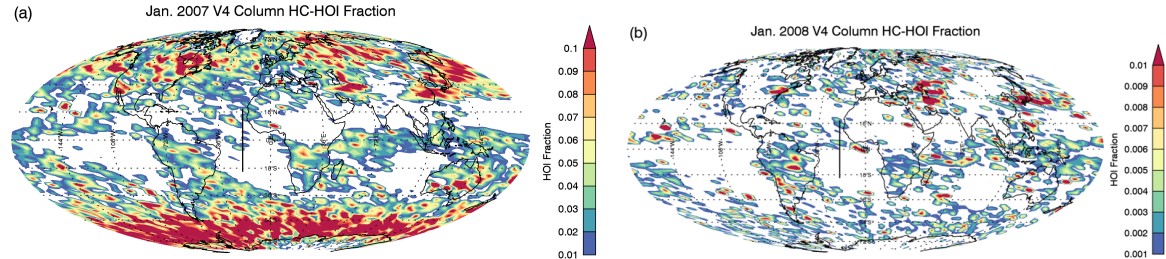

**Figure 7:** Comparison of the high-confidence HOI column fraction of cloudy bins observed at 0.3° and 3° by CALIOP during January 2007 and 2008, Panels (a) and (b), respectively. The column HOI fraction is calculated as the ratio of high-confidence HOI identified by the phase algorithm to the total number of cloudy bins observed by CALIOP at 5-80 km resolution, compiled at 2° latitude x 5° longitude, daytime and nighttime

10   together. The scale in Panel (b) is 1/10 of that in Panel (a).

The comparison between these two Januarys viewed at 0.3° and at 3° shows that the HOI volume fraction of observed clouds has been reduced by an order of magnitude, and that the distribution of column HOI fraction in January 2008 is instead more randomly distributed.

15  **3.3 Viewing angle impact on cloud optical depths**

A time series of zonal-mean ice cloud optical depths shown in Fig. 8 shows that the permanent CALIOP viewing angle change in November 2007 from 0.3° to 3° did not significantly impact the climate data record of cloud optical depths. The time series was constructed using all V4 high-confidence ROI detected between 1-10 km and from 60° S to 60° N latitude from June 2007–June 2009.

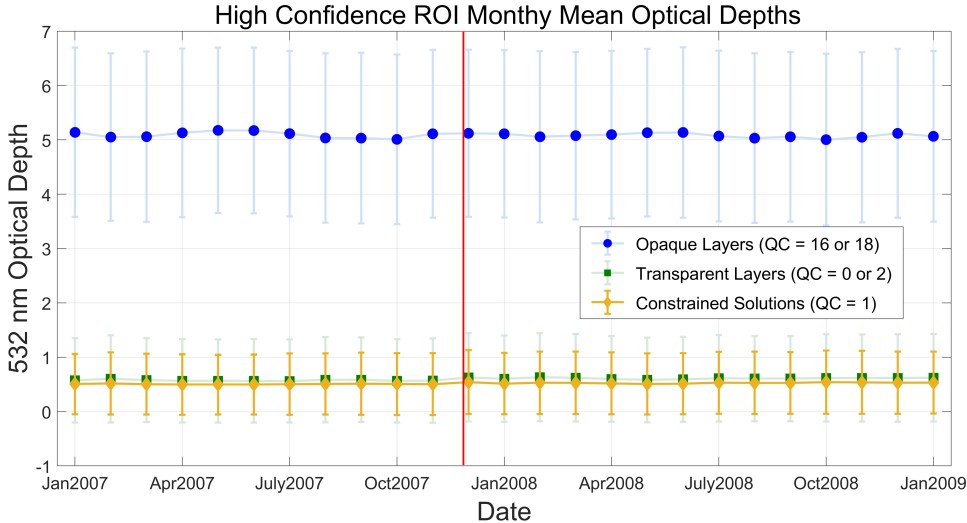





**Figure 8:** Monthly means for V4 high-confidence ROI layers averaged between 1-10 km and 60° S-60° N, before and after the viewing angle was changed permanently in November 2007. This viewing angle change is marked by the vertical red line. The layers are separated into extinction coefficient solution type (QC=0,1,2,16 or 18), with layers that completely attenuate the lidar beam in blue, layers constrained by two-way transmittance measurements in gold, and other layers that are transparent to CALIOP in green.

The transition from 0.3° to a 3° viewing angle highlighted by the vertical red line in Fig. 8 is not obvious in the retrieved optical thicknesses for high-confidence ROI, which was initially a concern when the viewing angle was changed. Artifacts in the time series would be most prominent in the unconstrained solutions of transparent ROI, where the lidar ratio is assigned based solely on cloud phase classification and centroid temperature (Young et al., 2018). Additionally, this shows that the extinction retrievals in totally attenuating clouds that are opaque to the lidar are working consistently at both viewing angles, with effective lidar ratios

constrained by $\gamma'_{532}$.

### 3.4 Evaluation of the CALIOP V4 phase algorithm with IIR microphysical indices

The CALIPSO Imaging Infrared Radiometer (IIR) measures radiances in three medium resolution channels centered at 8.65, 10.6 and 12.05 μm in the Earth's IR water vapor window. Measurements are taken in a 64 km x 64 km swath with a pixel size of 1 km, the center of which is aligned with the smaller CALIOP footprint (Garnier et al., 2018). The V4 IIR level 2 algorithm uses CALIOP

5 km layer products such as the number of layers detected and the 532 nm cloud layer centroid temperatures to select suitable scenes for retrievals of cloud effective emissivity and subsequent effective absorption optical thickness (Garnier et al., 2012). These retrievals are independent from CALIOP ice/water classification and can be used to evaluate the CALIOP cloud thermodynamic phase.

The effective microphysical index ($\beta_{eff}$ 12/10) is defined as the ratio of the effective absorption optical thicknesses in the 12.05 μm

and 10.6 μm channels. $\beta_{eff}$ 12/10 is very sensitive to particles of maximum dimension smaller than 60 μm (Mitchell et al., 2010). For temperatures between -38 °C and 0 °C, ice clouds and supercooled water clouds exhibit different IR signatures, with $\beta_{eff}$ 12/10 being unambiguously larger for water clouds than for ice clouds (Giraud et al., 2001; Mitchell and d'Entremont, 2012). Specifically, IIR $\beta_{eff}$ 12/10 is larger than about 1.2 for effective diameters smaller than about 25 μm (Fig.3(a) in Garnier et al., 2013) regardless of ice crystal shape or cloud phase, and slowly decreases towards about 1 as effective diameter increases. Figure 9(a) shows V4

IIR $\beta_{eff}$ 12/10 for CALIOP V4 high-confidence ROI clouds and water clouds, distributed by CALIOP layer $\gamma'_{532}$ and $\delta_v$, corresponding to a Hu phase diagram. These results are for single-layer cloud scenes over oceans between 60° S and 60° N, during January, April, July and October of 2008. All layers are transparent to the lidar and have 12.05 μm emissivities that are larger than 0.05 to prevent large IIR retrieval uncertainties in optically thin clouds.



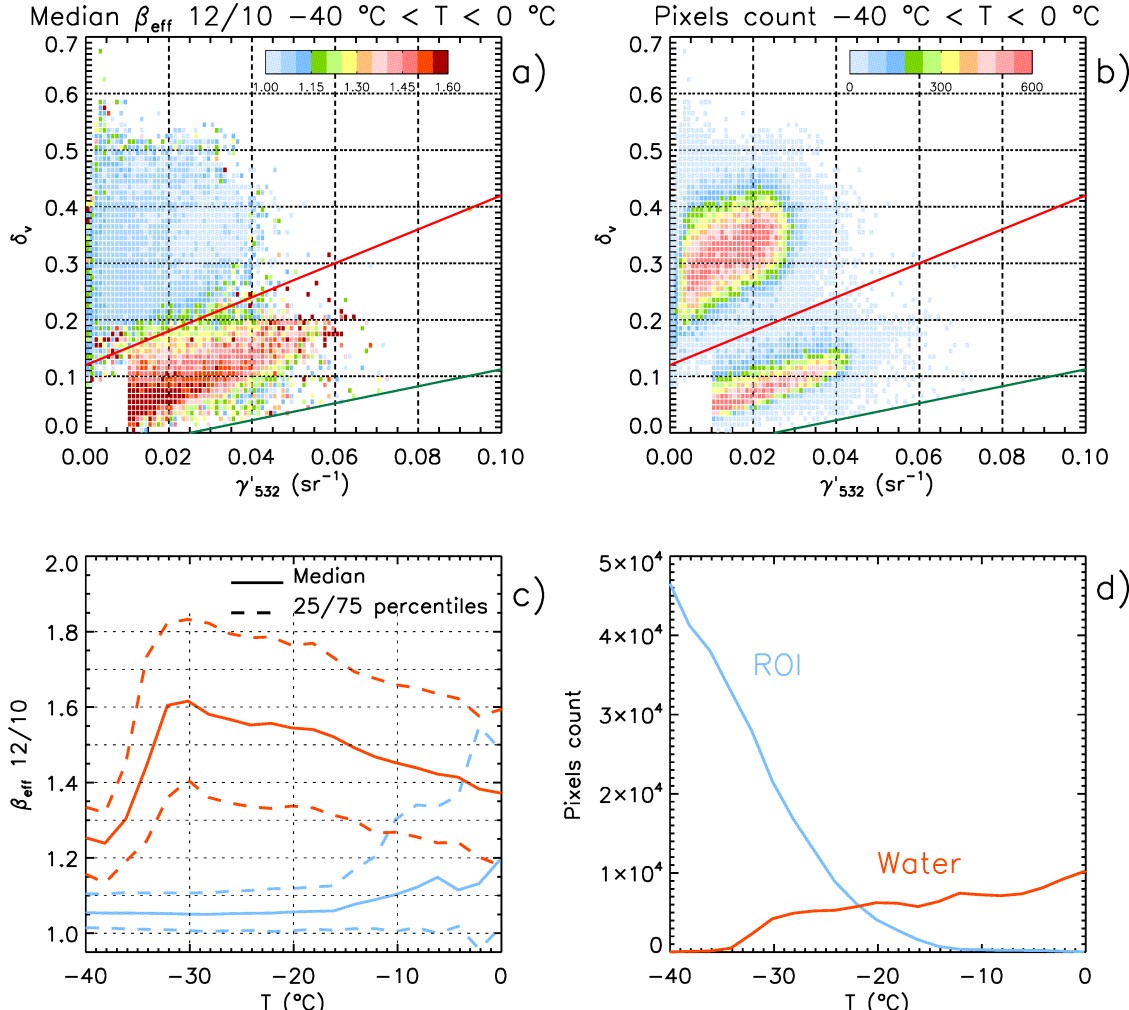

**Figure 9:** CALIPSO IIR 12/10 effective microphysical indices ($\beta_{eff}$ 12/10) for single-layer cloud segments that are transparent to CALIOP, between 60 degrees S and 60 degrees N over oceans during January, April, July and October 2008. Results shown are for 12.05 μm emissivities larger than 0.05. Panel (a) shows the distribution of $\beta_{eff}$ 12/10 by CALIOP integrated layer backscatter ($\gamma'_{532}$) and volume depolarization ratio ($\delta_v$). Panel (b) provides the corresponding number of analyzed 1 km IIR cloud samples. Panel (c) shows the median IIR $\beta_{eff}$ 12/10 as a function of CALIOP centroid temperatures between 0 and -40 °C, with CALIOP high-confidence water layers in blue and ROI layers in red. Panel (d) shows the distribution of the corresponding IIR pixel count.

The IIR $\beta_{eff}$ 12/10 is much larger for clouds in the Hu diagram water sector, with median values larger than 1.25, thereby indicating smaller effective diameters for water clouds as expected. In the ROI sector $\beta_{eff}$ 12/10 is almost universally less than 1.15, except for a small scattering of layers with either warm centroid temperatures or small $\gamma'_{532}$ that likely appear in the ROI sector because they are plotted by $\delta_v$ instead of $\delta_p$. These results demonstrate consistency between independent IIR microphysical index retrievals and the separation of ice and water clouds by the V4 CALIOP phase algorithm. The CALIOP layer depolarization and backscatter measurements add integrity to cloud phase determinations beyond using only a temperature-based ice/water climatology.





**4.0 Global and regional characterization of V4 vs V3 cloud phase differences**

**4.1 Impact of CALIOP cloud algorithm changes**

Changes made between V3 and V4 extend all the way back to improving both the 532 nm nighttime and daytime calibration, as well as re-calibration of the 1064 nm channel. These V4 calibration changes are documented extensively in a series of papers by
Kar et al., 2018; Getzewich et al., 2018; and Vaughan et al., 2019a. Since the phase algorithm makes use of all three CALIOP channels to make thermodynamic cloud phase decisions, changes to these calibrations in V4 can impact the distribution of assigned thermodynamic cloud phases by creating subtle changes in the $\gamma'_{532}$–$\delta_{p,eff}$ relationships. Changes to the CAD algorithm (Liu et al., 2019) and the surface detection algorithm (Vaughan et al., 2019b) also can impact the cloud phase assignment, in addition to changes made to the phase algorithm for V4 that are documented here. In a general sense, the impact of the CALIOP algorithm
changes on the cloud phase assignment can be understood by noting the following points:

- The improved 532 nm calibrations create enhanced sensitivity for layer detection, so there are more cloud layers in V4, and in many cases the cloud boundaries have been extended.
- The improved surface detection capability also enhances the layer detection capability, especially for low-level layers lying beneath layers that were identified as opaque in V3.
- The CAD algorithm has changed substantially and is now more aggressively identifying aerosol layers in the free troposphere.
- The CAD algorithm is now also applied to features in the stratosphere.
- V4 includes many more layers with both high and low CAD scores, although there are fewer CAD scores of 100 in V4. The increase in low CAD scores causes a large enhancement to the number of cloud layers assigned with an "unknown
phase".
- Cloud phase assignments are now made for single-shot and 1 km cloud layer detections.

**4.2 Assignment of high-confidence cloud phases**

Figure 10 illustrates the cumulative impact to the phase assignment of 5km cloud layer segments caused by changes to the upstream algorithms, combined with changes to the phase algorithm.

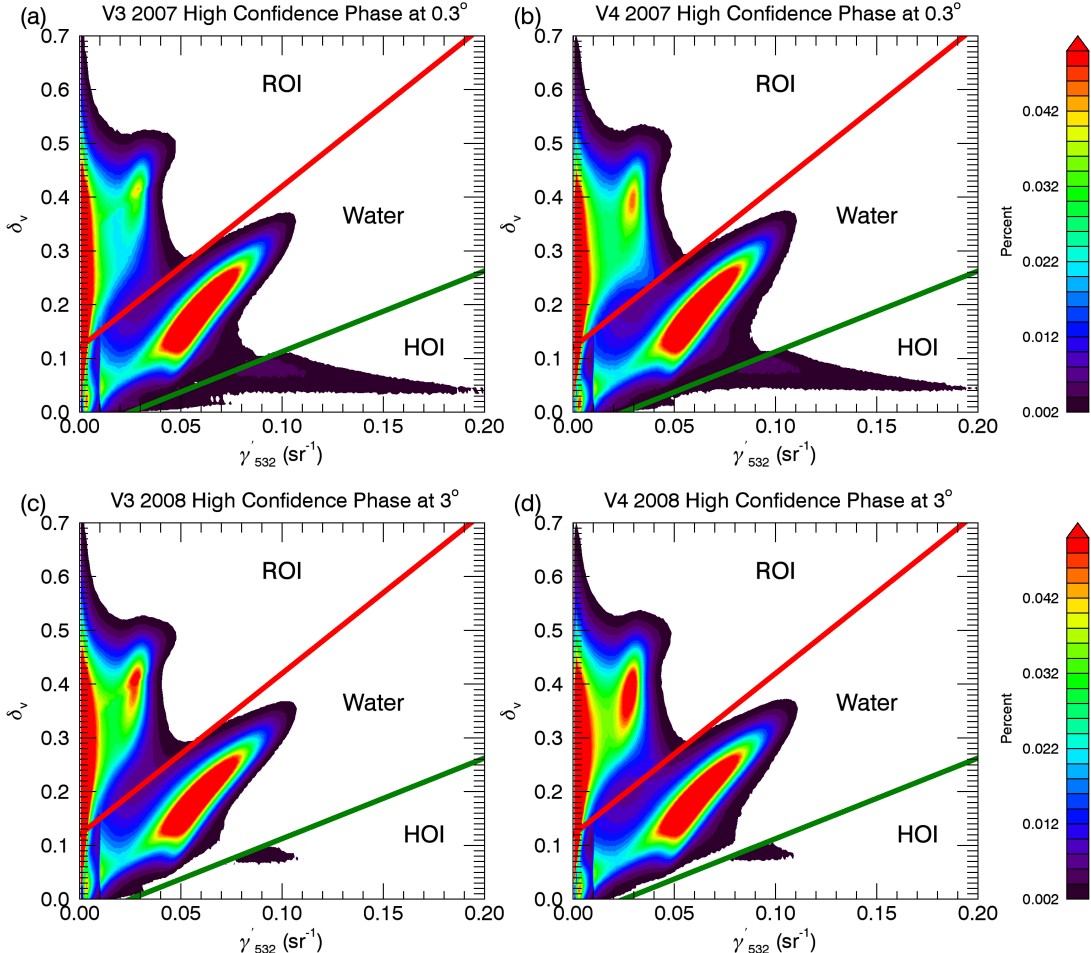

Figure 10: Hu phase diagrams showing the distribution of V3 and V4 $\delta_v$ vs. $\gamma'_{532}$ for all cloud layers with high-confidence cloud phase assignments detected at 5-80 km averaging intervals in 2007 and 2008. Combined daytime and nighttime $\delta_v$ vs. $\gamma'_{532}$ are 5 km cloud layer segments from the V3 and V4 5km cloud layer products.

The V4 phase algorithm is "highly confident" about layers detected at horizontal averaging lengths of 5km or longer more than 86% of the time, so Fig. 8 shows the majority of the V3-V4 layers. When comparing cloud phases between V4 and V3, it is important to note that some layer placements have changed relative to the Hu thresholds because layer $\gamma'_{532}$ in V4 has generally increased due to improved calibration and extended layer boundaries, while $\delta_v$ has stayed about the same. As previously mentioned in Section 4.1, there are also more cloud layers identified in V4 than in V3. There is a discontinuity in the distributions at $\gamma'_{532} = 0.01$ evident in the Fig. 10 water sectors. This discontinuity is caused by the switch from 532 nm volume depolarization to estimated particulate depolarization using the 1064 channel. The new 1064 calibration is much improved, and this causes changes in the estimated particulate depolarization for cloud layers with $\gamma'_{532} < 0.01$ sr$^{-1}$. The V4 increase in low CAD scores creates a large population of cloud layers with unassigned cloud phase that are not shown in Fig. 10. In V4, layers with "phase unknown" make up about 11% of layers detected at 5-80 km, and cloud fringes are 1.5%. Mid-confidence and low-confidence layers that are





assigned phases contrary to their sector on a Hu diagram based on temperature make up only about 1% of layers detected at a 5-80 km averaging interval.

### 4.3 V3/V4 case study comparison

5 In this section we show a case study to illustrate some of the major differences in cloud phase assignment that CALIOP data users who are used to using V3 can expect to see when switching to the new V4 data. The case study chosen is a nighttime overpass of "Superstorm Sandy" that occurred on 29 Oct. 2012 at about 07:15 Z. Sandy was a very large storm system with a combined tropical hurricane and extratropical mid-latitude cyclone (Kunz, 2013). This storm provides a large dynamic range of backscatter and depolarization, desirable for a meaningful comparison. Figure 11 shows a VIIRS day/night band image of Sandy with the CALIPSO overpass, and then from the west with the corresponding CALIOP backscatter image superimposed on top of the VIIRS

10 image.

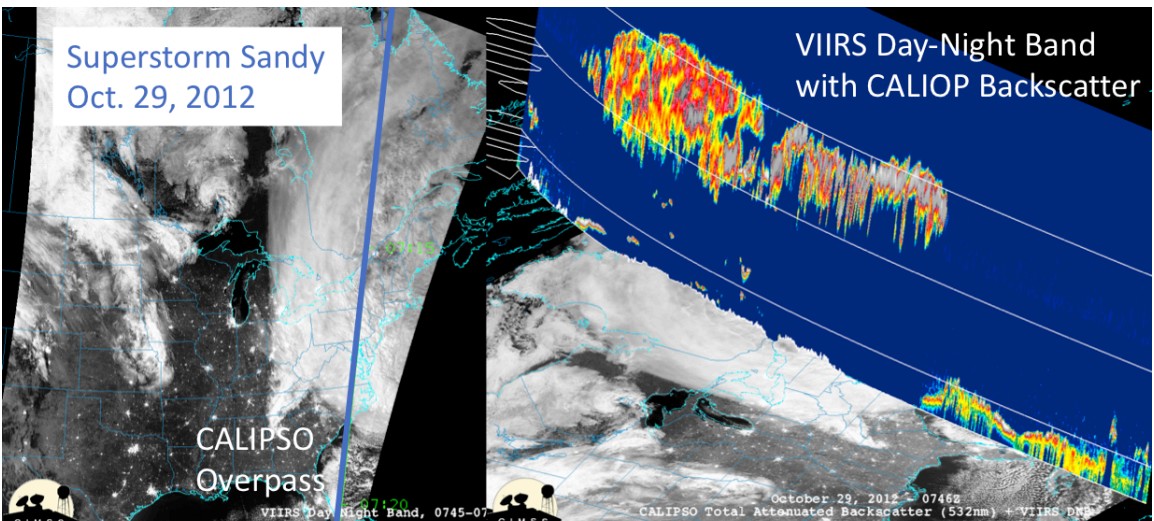

**Figure 11:** Overview of "Superstorm" Sandy, with VIIRS day/night image and CALIPSO Sandy overpass on the left-hand side, and then on the right-hand side is a view from the west, with the corresponding CALIOP backscatter browse image superimposed. This image was provided by William Straka III, at the University of Wisconsin.




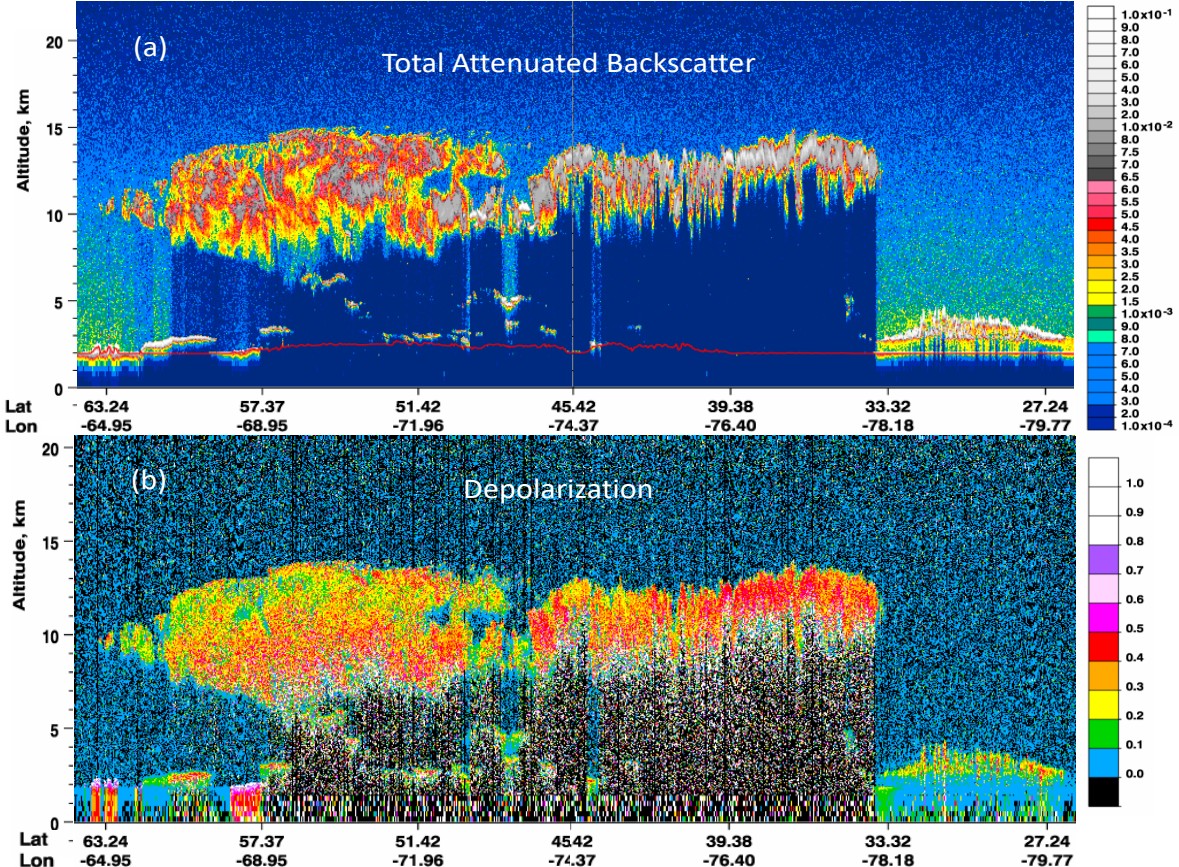

**Figure 12:** V4 CALIOP attenuated backscatter and depolarization browse images showing the CALIPSO nighttime overpass of Superstorm Sandy. at ~07:15 Z on Oct. 29, 2012.

Fig. 12 shows CALIOP browse images of the attenuated backscatter (a) and depolarization (b) from this nighttime overpass of

Sandy. The transition between tropical and midlatitude storms can be seen between about 47° N and 49° N. North of this transition

region the ice cloud is more diffuse and depolarizes less; south of this the ice clouds are optically thicker and depolarize more

strongly. Figure 13(a) shows the corresponding V3 thermodynamic phase assignment, with false identification of HOI (grey

stripes) at cold temperatures caused by natural variability or noise in $\gamma'_{532}$ and $\delta$. The absence of grey, HOI striping in Fig. 13(b)

shows the V4 correction of over-identification of HOI by confining the spatial coherence test to observations with $\gamma'_{532} > 0.02$ sr$^{-1}$

and only to water sector layers.

This case study in the storm transition region also shows how phases are chosen for geometrically thick cloud layers with mixed

water and ice phases. For reference, Fig13(c) shows the total backscatter coefficients and Fig. 13(d) shows the interpolated

MERRA-2 temperatures in the area outlined by the blue box in Fig. 13(b). Fig. 13(e) shows the cloud top, 532 nm centroid, mid-

cloud and cloud bottom temperatures in high-confidence water layers that occur in this region. These layers have $\delta_{p,eff}(\gamma'_{532})$ that

falls within the Hu diagram water sector. A few of these water layers have cloud tops at 12-13 km with temperatures around -60

°C but are still designated as water because they have a 532 nm backscatter centroid that is located towards the bottom of the cloud

layer at warmer temperatures. In V3 these layers would have been typed as ice clouds, based on the cloud mid-layer temperature,

and the same phase typing would also occur using cloud top temperature. While this happens only occasionally, cold-temperature



water bins in V4 can result from the assignment of only one thermodynamic phase to a geometrically deep layer with ice at the top of it, and water at the bottom. For layers with mixed ice and water phases, mis-classification of some bins within the layer is unavoidable without the capacity for a vertically-varied phase assignment.

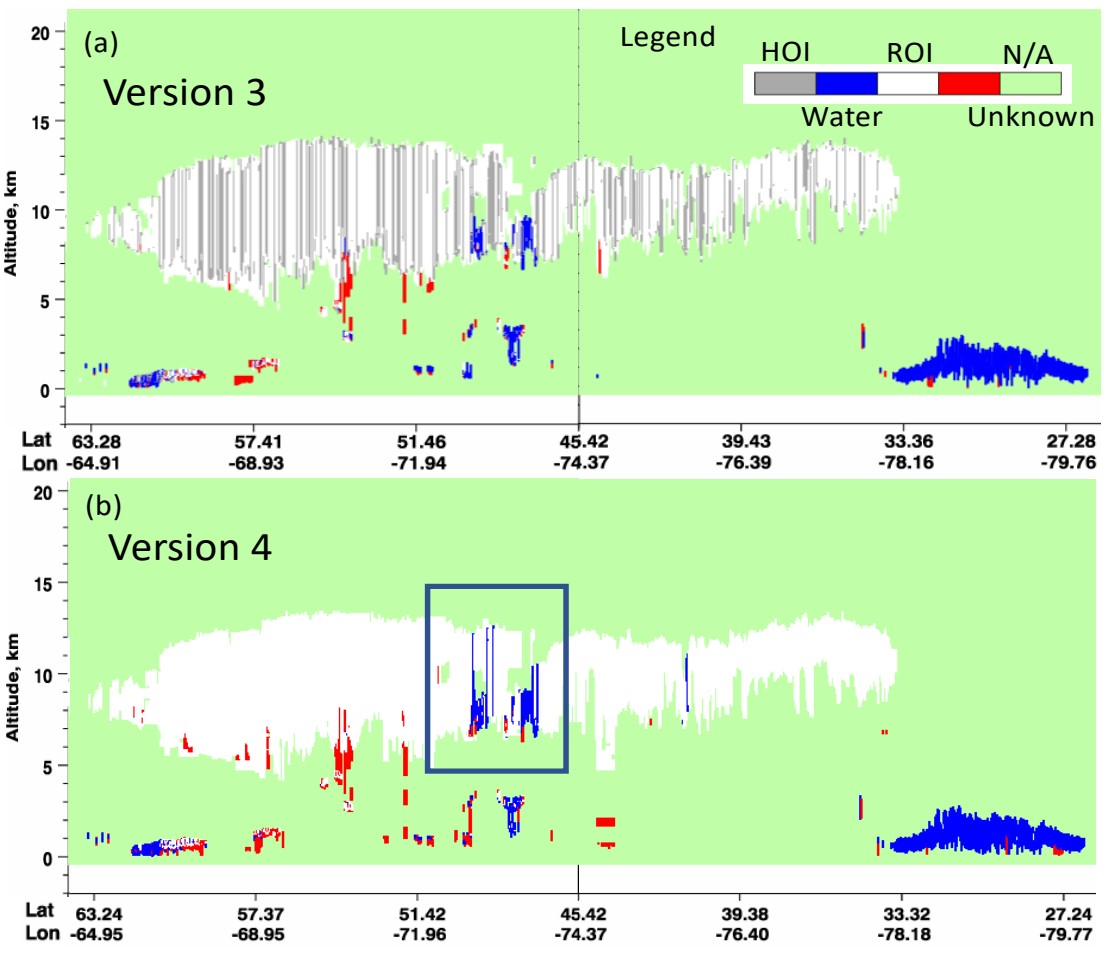

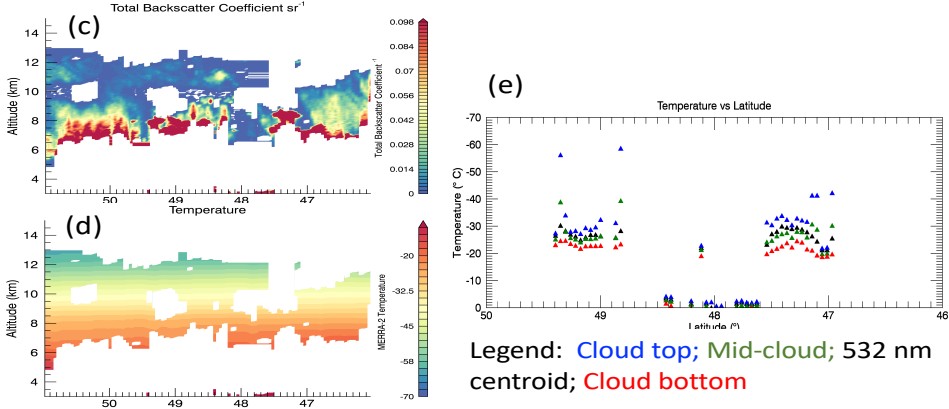





**Figure 13:** Comparison between V3.02 (a) and V4 (b) cloud thermodynamic phase assignments in the nighttime overpass of Sandy on Oct. 29, 2012. The legend shows unknowns (red), randomly-oriented ice ROI (white), water (blue) and "horizontally-oriented" ice, HOI (grey). Panel (c) shows an expanded view of the total backscatter coefficient and Panel (d) the MERRA-2 temperature field, at the transition between the mid-latitude and the tropical sections of the storm (blue box on Panel(b)) Panel (e) shows the cloud top, mid-cloud, 532 centroid and cloud bottom
temperatures for high-confidence water layers in this region.

## 4.4 Global zonal cross-sections

Figure 14 shows how the various cloud phase assignments are distributed in a global cross-section. The global occurrence frequencies are accumulated for each cloud phase by counting cloudy CALIOP sample bins in the V3 and V4 5 km cloud profile products, daytime and nighttime together during October 2010, when CALIOP has a 3° near-nadir viewing angle. As mentioned

above, the combined population of medium-confidence ROI and medium and low-confidence water is 1% or less of cloudy bins identified globally, so these are not shown here.

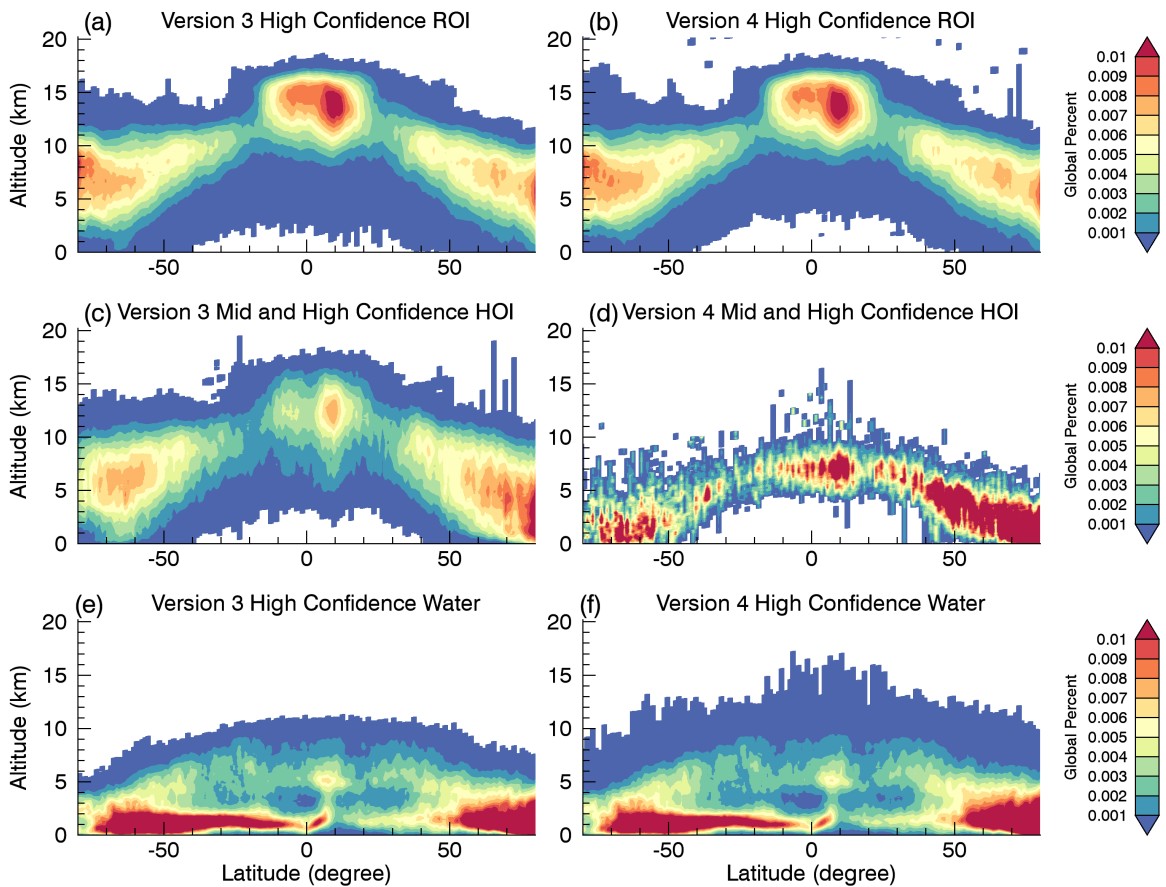

**Figure 14:** V3 (left-hand side) vs V4 (righthand side) comparison of normalized zonally-accumulated 60 m cloudy bins for daytime and nighttime
together in October, 2010. The analysis uses the 60 km cloud profile products and assumes that the topmost cloud phase below 8.2 km is correct for a 60m bin. Panels (a) and (b) compare high-confidence ROI, Panels (c) and (d) mid- and high-confidence HOI and Panels (e) and (f) show high-confidence water.

Aside from the HOI distributions, the overall pattern of ROI and water layers is very similar between V3 and V4. The striking

difference in HOI is caused by elimination of the spatial coherence test in V4, which was overly aggressive in identifying ice layers





as HOI in V3. As discussed earlier, this was likely occurring because the natural variability in the spatial correlations between $\gamma'_{532}-\delta_{p,eff}$ in clouds is larger than those caused by the greatly-reduced amount of specular reflections present in observations from the more oblique viewing angle.

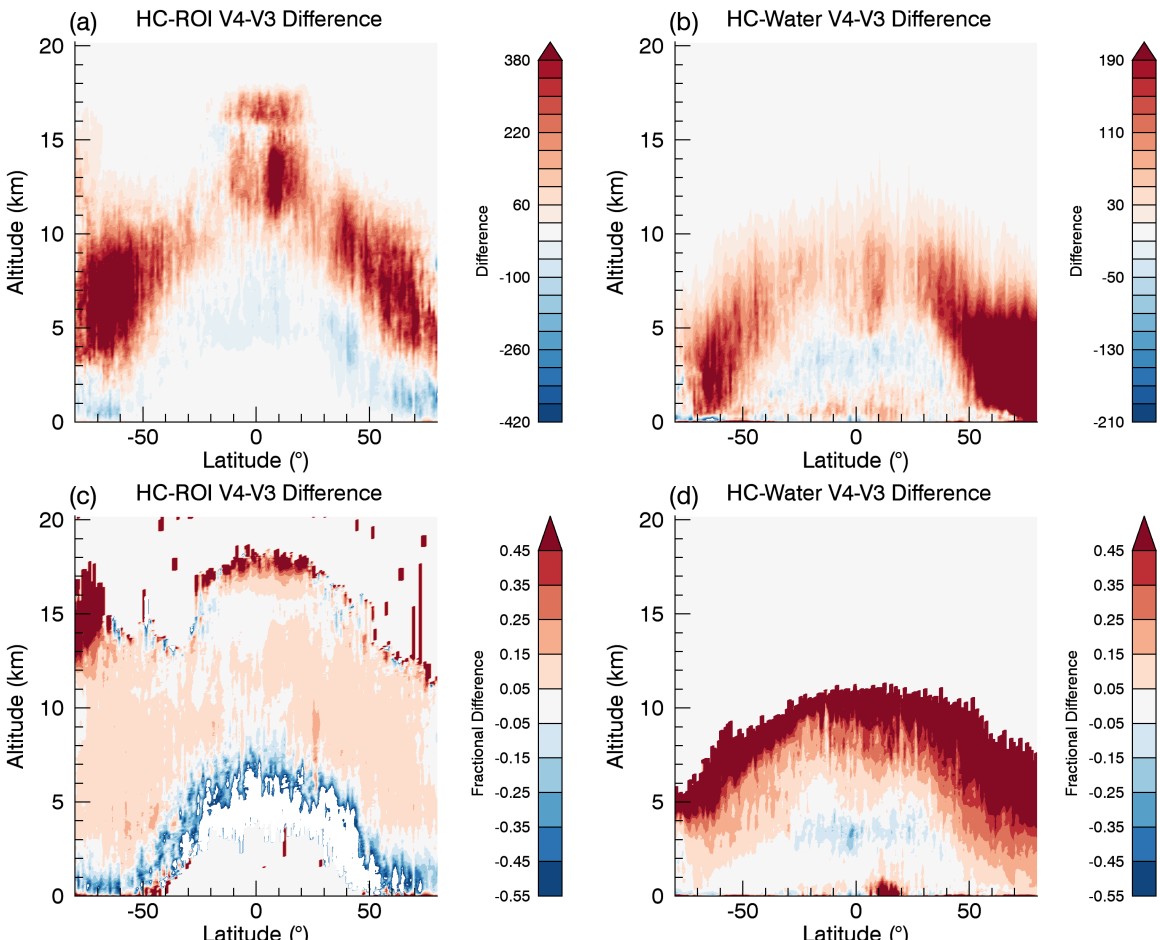

**Figure 15:** V4-V3 Difference plots for high-confidence ROI (HC-ROI, left-hand column) and high-confidence water (HC water, right-hand column) cloudy bins observed during October 2010. Panels (a) and (b) show the V4-V3 difference in accumulated cloudy bins for ROI and water and Panels(c) and (d) show the fractional V4-V3 difference when normalized to V4.

The subtler differences between V3 and V4 ROI and water are shown in the Fig. 15 difference plots. Fig.15(a) shows the changes in high-confidence ROI observations by differencing the accumulated occurrence of V4-V3 sample bins and Fig15(c) shows this difference normalized to V4. There are more ROI layers observed by CALIOP in V4 due to increased sensitivity from the V4 calibrations. In addition to increased ROI observations in the upper tropical troposphere, one can see the additional band of ROI located near the tropical tropopause at 16-18 km where features with tops above the apparent MERRA-2 tropopause are no longer assigned as "stratospheric features" and the CAD algorithm now identifies cirrus clouds. Some polar stratospheric clouds are also now designated as clouds by the CAD algorithm, evident in Fig. 15(c) poleward of 65° S. The ROI difference plot also shows the conversion of samples identified as HOI in V3 that are now ROI in V4. In V4 the larger signal at higher altitudes attenuates the lidar faster and so in Fig. 15(c) one can notice more ROI sample bins at lower altitudes in V3 than in V4 while there are more cloudy bins overall in V4.





Fig. 15(b) shows the difference in high-confidence water layers between V4 and V3. More water layers are now identified at high latitudes, poleward of 50° N. Fig. 15(d) shows clearly the V4 increase in water cloud samples at higher altitudes and decrease at lower altitudes, similar to what Fig. 15(c) shows for ROI. Using the centroid as opposed to the cloud top temperature explains the appearance of CALIOP low-temperature sample bins assigned as water, as shown in the case study from Superstorm Sandy. The

phase algorithm can assign only one phase for each cloud layer identified, including those that are geometrically thick with a large contrast between cloud top and cloud base temperatures. These cloud layers can have water or HOI at the bottom of them, which creates a large backscatter signal so that the layer centroid temperature is identified in this warmer part of the cloud. Alternatively, using only cloud top temperatures misses the water detected at the bottom of thick clouds. Other mixed-phase cloud layers may have small water droplets at the cloud top and precipitating ice crystals below them. Proper phase identification in either situation

would necessitate vertically differentiated phase assignments in mixed-phase cloud layers, which is highly desirable for future CALIOP phase algorithm improvement.

**4.3 V3-V4 contingency matrices**

Earlier sections have described the changes between the number of distinct cloud layers observed between V3 and V4. Cloud layers have varied horizontal detection lengths and naturally varying geometric vertical thicknesses, and so the atmospheric volume

that they occupy cannot be characterized by simply counting layers. To quantify how much the cloudy atmospheric volume changes between V3 and V4, the contingency matrices shown in Fig. 16 match the vertical 30m and 60m sample bins for all cloud phases, aerosol types and clear bins that have been identified as containing cloud or aerosol in either V3 or V4, using the vertical feature mask from the Level 2 cloud and aerosol profile products.

Fig. 16 shows the V3-V4 sample bin comparisons from 67deg N – 67deg S and from the surface to 20.2 km for both viewing angles, with daytime and nighttime shown separately. Sample bins that are clear in both V3 and V4 are suppressed because these would dominate the distribution. The vertical resolution is 30 m from 0 to 8.2 km and 60 m from 8.2 km to 20.2 km. These matrices do not show observations above 20.2 km because 20.2–30 km is in the stratosphere and was not included in the V3 L2 cloud and aerosol profiles. In this comparison we did not include any data poleward of the Arctic and Antarctic Circles to avoid the seasonal

differences between the 24-hour polar day and night, and to eliminate many polar stratospheric clouds from the comparisons.

Our comparison includes aerosol types because some aerosols, such as dust, depolarize the lidar return signal, causing about 5% V3-V4 cross-over between CALIOP observations of dusty aerosol layers and clouds (Liu et al., 2019). V4 aerosol types are described in detail in Kim et al., 2018, with additional insight into V3-V4 aerosol type changes. In this analysis aerosols are grouped

into dusty, other tropospheric and stratospheric. "Dusty" aerosol types include dust, polluted dust and the new dusty marine type for V4. Other tropospheric aerosols include marine, clean continental, polluted continental, smoke and elevated smoke. Stratospheric aerosol is used here to designate the stratospheric feature type in V3 and the stratospheric aerosol type in V4.

The diagonal on the comparison matrices represents the fraction of bins identified the same way in both V3 and V4, with the lower

righthand corner showing the total percent of matched V3-V4 bins with unchanged classifications. Otherwise, the right-hand column and the bottom row show the total percent of bins of each cloud and aerosol type observed in V3 and V4, respectively, and to within a small occasional rounding error, these add up to 100% for both. Since the V4 calibration and surface detection changes altered the boundaries of observed features, the conversion of clear bins to cloudy bins or aerosol can be substantial.





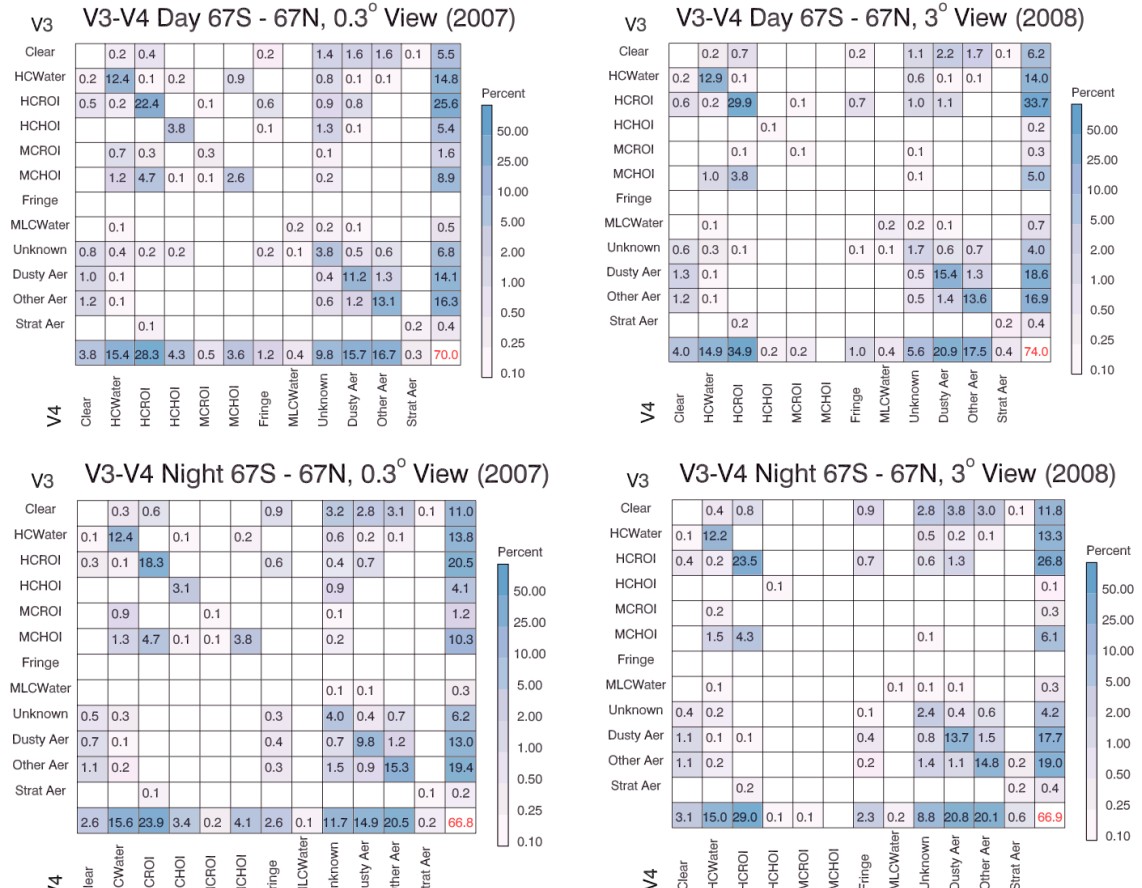

**Figure 16:** These panels show matched CALIOP V3 and V4 vertical sample bins from 67° N-67° S and from 0 to 20.2 km during 2007 at a 0.3° near-nadir viewing angle, and at 3° in 2008, with daytime and nighttime shown separately. Five km cloud profile files were matched for the first 100 daytime and nighttime granules during each month of each year, providing about 10 million matched sample bins. A sensitivity test demonstrated that including more granules during each month did not impact the results more than 1-2 tenths of a percent. The vertical resolution of sample bins shown here is 30 m from 0 – 8.2 km and 60 m from 8.2-20.2 km.  Month-to-month variability during each year was less than 2-3%, so we show only the combined plots here. These matrices show the percentages of sample bins that are identified as part of a feature in either V3 or V4, with clear/clear matches suppressed. V3 cloud phases and aerosol types occur along the rows, with corresponding V4 features along the columns. Aerosol types are grouped together here as "Dusty", "Other" and "Stratospheric".  Please see the text for further explanation.

These matrices contain a wealth of detailed information about the cumulative impact of V4 changes.  In both the 0.3° and 3° data, at night about 66-67% of sample bins remain the same type of feature in V4 as they were in V3. During the daytime there are even more unchanged bins, 70-74%. A major V3-V4 difference is the increase in high-confidence ROI in V4 3° data, partially due to the elimination of spatial coherence misidentification of medium-confidence HOI in optically thin ice clouds at both viewing angles, and with some additional V3 clear bins identified as ROI in V4 because of enhanced V4 calibration accuracy. As noted for layers in Section 2, another major change for V4 is that total HOI observed decreases from 7-8% at 0.3° to 0.1-0.2% at 3°, which is expected. The "unknown" type refers to the "unknown cloud phase" assignment, which often occurs when the CAD score is low, indicating ambiguity in the cloud-aerosol discrimination. Since the CAD scores are distributed differently in V4 than they are in V3 (Liu et al., 2019), there are many more "unknown" cloudy bins with low CAD scores in V4, and these are distributed between samples that were both aerosol and cloud features in V3.  There are more unknowns in the V4 0.3° data than in the 3° data, especially during the day.





There is a large overlap between high-confidence water identified in V3 and V4, with a small amount of V3 water samples becoming "unknown" or clear in V4, and additional water in V4 coming from bins that were clear in V3, indicating changes in cloud layer boundaries. Some additional water in V4 comes from sample bins that were medium-confidence ROI and HOI in V3.

There is also a small amount of cross-over between V3 aerosol layers and V4 water. The contribution of V3 aerosols to V4 ROI is less than 1%, but slightly larger in the 0.3° data than in the 3° data. However, in V4 between 1-2% of dusty aerosol layers were identified as ROI in V3, and this is a larger fraction in the 3° observations. It is worth noting that when multiple features exist in a column, CALIOP can detect underlying features provided the topmost layers are transparent to the lidar beams, which is a unique feature of lidar when compared with passive cloud detection. However, the larger the overhead attenuation becomes, the more

uncertain CAD and cloud phase determination also becomes. Therefore, the top-most layer assignments have the smallest amount of uncertainty.

Cloud "fringes", a new class in V4 not determined by the CAD or cloud phase algorithms, also include small amounts of V3 dust, polluted dust and smoke ("other aerosols"), although most V4 cloud fringes were identified as either clear air or ROI in V3.

Stratospheric aerosols are entirely new in V4, due to application of the CAD algorithm above the tropopause and the associated addition of new stratospheric aerosol types, described in Kim et al., 2018. V4 uses the MERRA-2 reanalysis tropopause height, which can impact whether some CALIOP range bins are apparently located in the troposphere or the stratosphere. The cross-over between V3 stratospheric aerosols and ROI clouds is still minimal, less than 1% of the total atmospheric cloudy volume.

## 5. Summary

**Impact of the near-nadir viewing angle on observations of specular reflections from oriented ice crystal faces**

Specular reflections from smooth ice crystal faces oriented perpendicular to the lidar beam impact both the total backscatter and the depolarization of the Clouds and Aerosol Lidar with Orthogonal Polarization (CALIOP) elastic backscatter lidar signal. These reflections are observed by CALIOP as expected from theory, from POLarization and Directionality of the Earth's Reflectances (POLDER) measurements and from ground-based and airborne lidar measurements. As predicted by POLDER, the impact of these

specular reflections on atmospheric measurements diminishes as the nadir viewing angle is increased beyond 1° and reduces by an order of magnitude at 3°. Changing the CALIOP viewing angle to 3° influences the observation of horizontally oriented plates and limits their observation in the atmospheric data to a very small fraction of observed ice clouds. Horizontally oriented ice (HOI) from smooth hexagonal faces of aggregates, columns or lofted plates in convection at higher altitudes may be more difficult to eliminate completely from the atmospheric data by adjusting the nadir viewing angle, but they also occur as only 0.5% of the

observations at 3°. There is an observable change in the mean backscatter and depolarization observations at near-nadir viewing angles of 1° or larger for layers with $\gamma'_{532} > 0.02$ sr$^{-1}$ between 0 and -40° C, with a reduction of 50% in 532 nm backscatter and increase of a factor of 5 in mean layer depolarization at the -15° C peak of the global HOI distribution.

### 5.2 What is the impact of the cloud phase algorithm changes?

There are significant changes in the distribution of Version 4 (V4) CALIOP cloud phase assignments from Version 3 (V3) and

earlier versions of the CALIOP data. These changes are the cumulative result of upgrading to the Modern Era-Retrospective-Analysis for Research and Applications-2 (MERRA-2) reanalysis for the temperature field, increased V4 calibration accuracy, an improved Cloud-Aerosol Discrimination (CAD) algorithm, overhead extinction retrievals and extensive changes to the cloud phase





algorithm. More ice and water cloud layers are identified in V4 than in V3, due to enhanced cloud layer detection resulting from better surface detection and more accurate attenuated backscatter measurement. The high phase confidence cloudy volume detected at 5-80 km increased by 5-9% between V3 and V4, calculated using cloudy 60 m vertical profile bins during the same daytime and nighttime 2007 and 2008 granules used for the comparison matrices shown in Figure 16. The larger end of this range occurs in

the data observed at 0.3°, and the smaller at 3°. When including unknown cloud phase layers and cloud fringes, the V3 to V4 cloud volume increase is 1-2%. Most of the medium-confidence HOI detected at 3° and half of that detected at 0.3° in V3 becomes randomly oriented ice or water in V4. The distribution of $\gamma'_{532}$ and $\delta_{p,eff}$ for cloud layers observed at 3° in V4 is dominated by a large population in the randomly oriented ice (ROI) sector, and so the spatial continuity test is only preserved for 0.3° data in the water sector, and for layers with $\gamma'_{532} > 0.02$ sr$^{-1}$.

The switch to evaluating cloud layer temperatures at the 532 nm backscatter centroid as opposed to the cloud top ensures that cloud layers are classified according to where the largest particle density occurs, increasing the accuracy of phase decisions for most cloudy bins. Occasionally this leads to the identification of the cold cloud tops of geometrically thick clouds as water, because of the dominance of backscatter by water at the warmer bottoms of cloud layers that contains both ice and water. As mentioned above,

specular reflections from HOI at the bottom of thick cloud layers are observed at 0.3°, but only rarely at 3°. The V4 algorithm must decide between water and ice in thick cloud layers with mixed cloud phases, and where this choice is ambiguous it now chooses based on where the backscatter is largest. Due to extinction algorithm improvements combined with improved phase detection accuracy, a time series of mean cloud optical depth does not show a significant discontinuity across the November 2007 viewing angle change from 0.3° to 3°.

Contingency matrices show that cloud edges have changed in V4, and that consequently the total atmospheric volume containing clouds in V4 has increased relative to V3, as mentioned above. Cloud fringes that were identified by the CAD algorithm as aerosol layers were subsequently found by spatial proximity to occur at the edges of ice clouds and are identified as no-confidence ROI in V4. These cloud fringes were mainly either clear air or ROI in V3, although there is a small population that was classified as

aerosol in V3. There are more cloud layers with unknown phase in V4, mainly due to CAD algorithm changes, but also due to reduced SNRs in some of the thin cloud layers that can now be identified in V4 and from negative effective particulate depolarization calculated in the HOI phase sector.

### 5.3 Where does the phase algorithm excel, and where is it challenged?

The cloud phase algorithm is most accurate where there is little overhead 532 nm signal attenuation, when a cloud layer is optically

thick enough to provide a robust signal in both the parallel and perpendicular 532 nm channels and composed of one single thermodynamic phase, either ice or water. For clouds with centroid temperatures colder than -40 °C, the cloud phase is identified as randomly oriented ice with high confidence 95-100 % of the time. At warmer temperatures the cloud phase differentiation between water and ice is usually more accurate when $\gamma'_{532}$ is larger than 0.01 sr$^{-1}$, because of the clear separation between water and ice populations in correlations between the $\delta_v$ and $\gamma'_{532}$. In the 0 to -40 °C temperature range we showed an excellent consistency

between independent IIR microphysical index retrievals and the CALIOP separation of high-confidence ROI and water for single cloud layers.

On the other hand, thin cloud layers ($\gamma'_{532} < 0.01$ sr$^{-1}$), or layers with a large amount of overhead signal attenuation cause difficulty because of low 532 nm signal to noise ratios, and uncertainty in correcting for an accurate amount of overhead optical depth. Since



the particulate depolarization is estimated using the 1064 channel for thin cloud layers, attenuation of the 1064 channel can impact the phase determination for these thin cloud layers as well. As discussed in the previous section, the phase algorithm is also challenged by geometrically thick cloud layers that may contain both ice and water, in which case a choice between phase assignments has to be made. In V4 the phase algorithm chooses the cloud phase occurring at the 532 nm attenuated backscatter

centroid where the most backscatter occurs.

## 6. Plain language summary

CALIOP data users will find that there are more cloud layers detected in V4, with edges that may extend further than in V3, for a combined increase in total atmospheric cloud volume of 6-9% for high-confidence cloud phases and 1-2% for all cloudy bins, including cloud fringes and unknown phases. In general, the increase between V3 and V4 cloudy bins is larger for data acquired

at 0.3° than it is for 3° data. In V4 there are also fewer cloud layers identified as HOI, particularly in the 3° observations. There is increased confidence in the V4 cloud phase assignments, due to enhanced V4 backscatter calibrations, a better CAD algorithm, new overhead extinction retrievals and our overhaul of the cloud phase algorithm. Where the cloud-aerosol discrimination is uncertain, indicated by a CAD score of less than 20, the cloud phase has been designated as "unknown". It is remarkable how well the dual-channel lidar technique, originally demonstrated by Schotland et al. in 1971, works for differentiating between water and

ice clouds in global observations from a space platform.

## 7. Data availability

CALIPSO data is available at the Atmospheric Science Data Center, and can be accessed by using this link: https://eosweb.larc.nasa.gov/project/calipso/calipso_table.

The following CALIPSO data products were used in this study:

Fig. 1: V4.10 CALIPSO level 2 5km cloud layer product (Vaughan et al., 2018; NASA Langley Research Center Atmospheric Science Data Center; https://doi.org/10.5067/CALIOP/CALIPSO/LID_L2-05kmCLay-Standard-V4-10; last access August, 2019);

Fig. 2: V4.20 CALIPSO level 2 5km cloud layer product (Vaughan et al., 2018; NASA Langley Research Center Atmospheric Science Data Center; https://doi.org/10.5067/CALIOP/CALIPSO/LID_L2-05kmCLay-Standard-V4-20; last access August, 2019);

Fig. 3: V4.20 CALIPSO level 2 5km cloud layer product, see Fig. 2;

Fig. 4: V3.01 CALIPSO level 2 5km cloud layer product (Vaughan et al., 2018; NASA Langley Research Center Atmospheric Science Data Center; https://doi.org/10.5067/CALIOP/CALIPSO/CAL_LID_L2-05kmCLay-ValStage1-V3-01_L2-003.01; last access August, 2019);

Fig. 5: N/A (flow diagram);

Fig. 6: V4.10 CALIPSO level 2 5km cloud profile product (Vaughan et al., 2018; NASA Langley Research Center Atmospheric Science Data Center; https://doi.org/10.5067/CALIOP/CALIPSO/CAL_LID_L2-Standard-V4-10_L2B-004.10; last access

August, 2019);

Fig. 7: V4.10 CALIPSO level 2 5km cloud profile product (see Fig. 6);

Fig. 8: V4.10 CALIPSO level 2 5km cloud layer product (see Fig. 1);


Fig. 9: CALIPSO level 2 IIR; In production (Nov. 2019); data used here is available from the authors upon request.



Fig. 10: V4.10 CALIPSO level 2 5km cloud layer product (see Fig. 1);

Fig. 11: VIIRS day/night band, image of Sandy provided by William Straka III of U. Wisconsin, CIMSS, and V3.02 CALIPSO level 1 profile product (Vaughan et al., 2018; NASA Langley Research Center Atmospheric Science Data Center; https://doi.org/10.5067/CALIOP/CALIPSO/CAL_LID_L1-Prov-V3-02_L1; last access August, 2019);

Fig. 12: V3.02 CALIPSO level 1 cloud profile product (see Fig. 11);

Fig. 13: V3.02 CALIPSO level 1 profile product (see Fig. 11); V4.10 CALIPSO level 1 profile product (Vaughan et al., 2018; NASA Langley Research Center Atmospheric Science Data Center; https://doi.org/10.5067/CALIOP/CALIPSO/CAL_LID_L1-Prov-V4-10_L1; last access August, 2019), V4.20 CALIPSO level 2 cloud profile product (Vaughan et al., 2018; NASA Langley Research Center Atmospheric Science Data Center; https://doi.org/10.5067/CALIOP/CALIPSO/CAL_LID_L2-Standard-V4-20_L2B-004.20; last access August, 2019) and V4.20 CALIPSO level 2 cloud profile product (see Fig. 2);

Fig. 14: V4.10 CALIPSO level 2 5km cloud profile product (see Fig. 6);

Fig. 15: V4.10 CALIPSO level 2 5km cloud profile product (see Fig. 6);

Fig. 16: V3.01 CALIPSO level 2 5km cloud profile product (see Fig. 4) and V4.20 CALIPSO level 2 5km cloud profile product (see Fig. 13);

The CALIPSO level 1 and level 2 data products are also available from the AERIS/ICARE Data and Services Center, (http://www.icare.univ-lille1.fr, AERIS/ICARE; last access: August. 2019).

**8. Supplement link**

*Provided by the editors.*

**9. Author contributions**

First author Melody Avery led the V4 CALIOP phase algorithm design study, and produced much of the analysis and figures as well as the first draft of this paper. Robert Ryan was responsible for writing the V4 phase algorithm production code, and Brian Getzewich wrote the original Version 2 and 3 phase algorithm code and consulted on the development of V4. As CALIPSO algorithm development lead and Principal Investigator, respectively, Mark Vaughan and David Winker made critical decisions regarding the V4 phase algorithm design and functionality. As the originator of the CALIOP phase algorithm, Yongxiang Hu provided insight into the theoretical basis underlying CALIOP phase thermodynamic assignments. Anne Garnier and Jacques Pelon provided the IIR microphysical index analysis. Carolus A. Verhappen directed the near-nadir viewing angle adjustments.

**10. Competing interests**

Author Jacques Pelon is a co-guest editor for the "CALIPSO Version 4 Algorithms and Data Products" special issue in *Atmospheric Measurements Techniques* but did not participate in any aspects of the editorial review of this manuscript. All other authors declare that they have no conflicts of interest.

**11. Special issue statement**

This article is part of the special issue "CALIPSO version 4 algorithms and data products". It is not affiliated with a conference.

**12. Acknowledgements**

We thank the NASA Science Mission Directorate, Earth Science Division for funding the CALIPSO project. Bryan Baum's thoughtful comments and suggestions caused us to generate a better manuscript. We thank the Aeris infrastructure and Nicolas Pascal for IIR code development support and for IIR data processing at the Aeris/ICARE Data and Services Center. William Straka III created the combined VIIRS/CALIOP images shown in Figure 11.



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
