# Peer review of "CALIOP V4 Cloud Thermodynamic Phase Assignment and the Impact of Near-Nadir Viewing Angles"

_Atmospheric Measurement Techniques, 2019_

## Referee Comment (RC1) · Anonymous Referee #1 · 23 Mar 2020

In my view, this is a strong manuscript that has a topic suitable for the journal and its special issue, and presents high-quality work relevant to a large community. From my perspective, the two key contributions are (a) a detailed description of the updated algorithm for cloud phase determination in the most recent version of the operational CALIOP cloud product, and (b) an analysis of cloud phase differences that occur when we switch from analyzing the previous version of CALIOP cloud products to analyzing the newest version. The first topic is important for all researchers analyzing CALIOP cloud phase data; the second topic is important for the wide community of people who have experience with the previous version of CALIOP data. The methodology is sound, and the presentation is clear. I do have a few suggestions for minor improvements in

wording or clarity, but none of my comments raise significant concerns.

Specific comments:

Page 3, Line 19: I suggest removing "the" from in front of "CALIOP".

Equation (5): Is the last relationship meant to be "equal", as opposed to "nearly equal"? If it is indeed meant to be "equal", it would help to explain why the equality.

Page 10, Line 5: It would help to add a sentence on explaining why the positive correlation for water clouds (I guess multiple scattering?) and why the negative correlation for ice clouds (perhaps due to larger backscatter being associated with a larger fraction of HOI, which have a smaller depolarization than ROI?). Of course, readers could always look up the cited reference for more information, but adding a sentence could make it more convenient and ensure that all readers understand the physical concept behind this spatial coherence test (which plays a significant role in the manuscript).

Page 10, Lines 26-28: I suggest clarifying whether it's the color scale that would be adjusted and mentioning that this adjustment is not shown in the figure. Most importantly, though, I recommend mentioning the appropriate correlation coefficient value in the text, or including it into the figure. This would help because, just looking at the current figure panels, it is not clear that the correlation is indeed positive in the water sector.

Page 20, Line 9: The words "the view" seem to be missing in front of "from the west".

Page 26, Lines 3 and 4 of the caption for Figure 16: It would help to explain what is meant by "granule". Just a few words saying that it's a roughly half-orbit extending from one terminator to the next could be sufficient. This word (and maybe the data selection used for Fig. 16) should also be explained in Section 5.2 (Line 4 of Page 28) for the sake of readers who jump to the conclusions right after reading the abstract, without going through the detailed text first.

Page 26, Lines 20-21: It could be interesting to include some thoughts on whether the increase in unknown phase is likely caused by the presence of unidentified HOI

(or perhaps by surface reflection?). The same applies to the increase in cloud volume (Page 28, Line 3), which is only 5% for 3° view angle, but is 9% for 0.3° view angle.

Page 27, Line 20: The section number 5.1 is missing from in front of the section title.

Page 27, Lines 27-30: I suggest adding something along the lines of "The impact of" at the beginning of this sentence, if I am right in assuming that it is not the crystals themselves that we try to eliminate (by excluding clouds with HOI from the CALIOP products), but we try to minimize their impact on the retrieved cloud properties. Also, it would help to delete, replace, or explain the word "more", as I was not sure about "more than what"? Finally, for the sake of clarity, I suggest replacing "they also occur as" in Line 29 by something like "this is a fairly minor issue, as HOI affect".

Page 28, Line 30: I suggest adding "is" in front of "composed".

---

## Referee Comment (RC2) · Anonymous Referee #2 · 22 Apr 2020

General Summary: Depolarization ratio of CALIOP 532 nm signal is an additional variable that helps discriminate cloud thermodynamic phase. Passive radiometers are subject to "ambiguities" that CALIOP can alleviate. There is a -15 °C peak in HOI occurrence globally, and testing with different view angles results in a decrease in depolarization ratio by a factor of 5, and the mean backscatter decreases by 50% when view angle is changed from 0.3° to 3°. The CALIOP cloud phase algorithm for Version 4 does not "over-identify" HOI at 3°. V4 has increased detection of cloud layers, and total cloud volume increased 6-9% over V3 for "high-confidence" determinations and by 1-2% for all determinations.

Review:

This paper is high-quality and pushes lidar remote sensing science forward significantly. It is written very well, and this reviewer has no major criticisms that need to be addressed prior to publication. However, there are a few minor corrections that should be addressed to ease in readability and clarity, and those are noted below.

Minor Comments:

Line 33-34: It should be stated, at least briefly, how IIR observations complement the cloud phase determinations.

Line 19: "HOI" has already been defined on Line 29 on Page 3.

Lines 10-15: It would be helpful to the reader to discuss a range of expected differences between the molecular-only layer attenuated backscatter and the total layer attenuated backscatter, so that the assumption that ðİŻ£<(ðİŚğ) âĽˇÈ ðİŻ£F(ðİŚğ) is clear.

Line 37: remove closing parenthesis after Mioche et al., 2010.

Line 22: Replace ", however this has been corrected in V4.5 and" with ". However, this has been corrected in V4.5, and"

Line 23-24: Rewrite to "If the centroid temperature is > 0 °C the layer will be assigned as low-confidence water, and occurs only rarely in < 0.1% of layers detected at 5-80 km."

Page 14 Line 15 (and throughout): Use consistent terminology. Either "viewing angle" or "view". Some readers may confuse "view" for "field of view" taken out of context.

Is the color scale for the CALIOP 532 nm backscatter curtain the standard one used in quicklooks (as in Figure 12a)? It appears that either the color scale is shifted or small values are being masked out, so please include a color bar or note about the values presented. There is no appearance of "total attenuation" underneath the clouds in Figure 11 as there is in Figure 12.

Line 20: Fix the notation of "67deg N – 67deg S" to a symbolic representation of degrees (As in the caption for Figure 16).

---

## Author Comment (AC1) · 20 May 2020

**Responses to Reviewer 1:**

**"CALIOP V4 Cloud Thermodynamic Phase Assignment and the Impact of Near-Nadir Viewing Angles"** *by* **Melody A. Avery et al.**

**Summary:**
The authors would like to thank two reviewers for carefully reading our paper. We appreciate the supportive comments made by both reviewers and we are happy to make the suggested minor changes. In each case we have made the suggested change, which improves the clarity of our presentation, so we are grateful to the reviewers. For ease of tracking these changes we have itemized our action for each change suggested. Our comments are in blue, after each comment.

**Reviewer 1:**

Page 3, Line 19: I suggest removing "the" from in front of "CALIOP".

> Have done this, although the acronym describes a lidar system the sentence now reads:

" This paper describes upgrades and changes to cloud phase and phase confidence assignments made using measurements from CALIOP, the first-ever space-based polarization-sensitive lidar."

Equation (5): Is the last relationship meant to be "equal", as opposed to "nearly equal"? If it is indeed meant to be "equal", it would help to explain why the equality.

> The equation, taken directly from Hu et al., 2009 might be confusing as written. It is an approximation for 532 nm particulate depolarization, as explained by the sentence beginning on line 12, but we call it $\delta_{1064}$ to avoid confusion with $\delta_v$, both approximations for $\delta_p$ that are used in the algorithm. So as originally written it is a definition. To simplify the expression, and to complement the text, Eq. 5 is now written:

$$\delta_{1064} \equiv \frac{\beta_{532,\perp}}{\beta_{1064,\parallel}} \approx \frac{\beta_{532,\perp}}{\beta_{1064} - \beta_{532,\perp}} = \frac{1}{\frac{\beta_{1064}}{\beta_{532,\perp}} - 1} \quad (5)$$

Page 10, Line 5: It would help to add a sentence on explaining why the positive correlation for water clouds (I guess multiple scattering?) and why the negative correlation for ice clouds (perhaps due to larger backscatter being associated with a larger fraction of HOI, which have a smaller depolarization than ROI?). Of course, readers could always look up the cited reference for more information but adding a sentence could make it more convenient and ensure that all readers understand the physical concept behind this spatial coherence test (which plays a significant role in the manuscript).

>To address this comment, we have added these two sentences starting on line 4:

"This test is based on the observation that due to multiple scattering, $\gamma'_{532}$ and $\delta_{p,eff}$ are positively correlated in water clouds. In contrast they are negatively correlated in HOI clouds because

specular reflections from HOI ice crystals in a cloud layer increase $\gamma'_{532}$ but do not depolarize the light."

Page 10, Lines 26-28: I suggest clarifying whether it's the color scale that would be adjusted and mentioning that this adjustment is not shown in the figure. Most importantly, though, I recommend mentioning the appropriate correlation coefficient value in the text or including it into the figure. This would help because, just looking at the current figure panels, it is not clear that the correlation is indeed positive in the water sector.

> Adding the correlation coefficient is a good idea, thanks to the reviewer for the suggestion. The last sentence of this paragraph now reads:

" Further, at 0.3° there is a weakly negative correlation of -0.07 between backscatter and depolarization in the water sector, in contrast to a positive correlation of 0.38 at 3°, indicating a significant misidentification of water as HOI when the spatial coherence test is applied at the larger viewing angle."

Page 20, Line 9: The words "the view" seem to be missing in front of "from the west".

> We added this to the text.

Page 26, Lines 3 and 4 of the caption for Figure 16: It would help to explain what is meant by "granule". Just a few words saying that it's a roughly half-orbit extending from one terminator to the next could be sufficient. This word (and maybe the data selection used for Fig. 16) should also be explained in Section 5.2 (Line 4 of Page 28) for the sake of readers who jump to the conclusions right after reading the abstract, without going through the detailed text first.

> This sentence has been added to p. 26, lines 3 and 4:

" CALIOP data granules extend from one terminator to the next, thus dividing each orbit into separate daytime and nighttime segments."

> The paragraph starting on line 6 on page 28 now states:

" Looking at the V3 to V4 changes in cloud volume necessitates using 60 m vertical profile bins. CALIOP data granules extend from one terminator to the next, thus dividing each orbit into separate daytime and nighttime segments. The first 100 daytime and nighttime granules in each month during 2007 and 2008 were used for the comparison matrices shown in Figure 16. The distributions did not change significantly when more granules were included. As determined from this ensemble of profile data the high phase confidence cloudy volume detected at 5-80 km increased by 5-9% between V3 and V4."

Page 26, Lines 20-21: It could be interesting to include some thoughts on whether the increase in unknown phase is likely caused by the presence of unidentified HOI (or perhaps by surface reflection?). The same applies to the increase in cloud volume (Page 28, Line 3), which is only 5% for 3° view angle, but is 9% for 0.3° view angle.

> This is indeed interesting to consider, and not trivial to explain since we can't compare observations from both viewing angles during the same time period, and there is some natural variability that may be occurring. When comparing unknowns from both viewing angles, missing HOI seems more likely to be found at 0.3°, based on the layer 532 nm centroid temperatures. We added this discussion to the cited paragraph (which now occurs on page 28):

" At 3° almost all unknowns that are not due to a low CAD score occur with centroid temperatures between 20 and -20 °C, and slightly more than half of these occur below 2 km in the tropics where the lidar signal is more likely to be significantly attenuated and the Level 2 algorithms have increased errors compounded by extinction due to overhead cloud and aerosol layers. At 0.3° about 3% of the additional "phase unknowns" may be HOI that have negative depolarization, as unknown layers are also observed at -20 to -40 °C and can have $\gamma'_{532} > 0.02$ sr$^{-1}$. It may be possible to recover more than half of these as HOI with the effective HOI depolarization correction included in Version 4.5."

While the cloud volume increases less in the 3° data, there are more dusty aerosols; so we added a sentence on page 29, lines 16-18:

" This is also reflected in the smaller increase in total cloud volume from V3 to V4 in the 3° data, which has 0.5% fewer water clouds and 1.5% fewer ice clouds, but also has 5% more dusty aerosol bins than at 0.3°."

Page 27, Line 20: The section number 5.1 is missing from in front of the section title.

> We added this, thank you for catching this mistake.

Page 27, Lines 27-30: I suggest adding something along the lines of "The impact of" at the beginning of this sentence, if I am right in assuming that it is not the crystals themselves that we try to eliminate (by excluding clouds with HOI from the CALIOP products), but we try to minimize their impact on the retrieved cloud properties. Also, it would help to delete, replace, or explain the word "more", as I was not sure about "more than what"? Finally, for the sake of clarity, I suggest replacing "they also occur as" in Line 29 by something like "this is a fairly minor issue, as HOI affect".

> This is a good point as we are not trying to eliminate the ice, but want to eliminate specular reflections from the ice, and the "more" needs to be explained. This is a good example of being too close to a problem to communicate well. The sentence has been changed to clarify both points:

"Specular reflections from horizontally oriented ice (HOI) from smooth hexagonal faces of aggregates, columns or lofted plates in convection at higher altitudes may be more difficult to eliminate completely from the atmospheric data than HOI in stratiform clouds by adjusting the nadir viewing angle because they do not align horizontally in strong updrafts, but they also occur as only 0.5% of the observations at 3°."

Page 28, Line 30: I suggest adding "is" in front of "composed".

> Done.

---

## Author Comment (AC2) · 20 May 2020

**Responses to Reviewer 2:**

**"CALIOP V4 Cloud Thermodynamic Phase Assignment and the Impact of Near-Nadir Viewing Angles"** *by* **Melody A. Avery et al.**

**Summary:**
The authors would like to thank two reviewers for carefully reading our paper. We appreciate the supportive comments made by both reviewers and we are happy to make the suggested minor changes. In each case we have made the suggested change, which improves the clarity of our presentation, so we are grateful to the reviewers. For ease of tracking these changes we have itemized our action for each change suggested. Our comments are in blue, after each comment.

**Reviewer 2:**

Line 33-34: It should be stated, at least briefly, how IIR observations complement the cloud phase determinations.

> This sentence was added after the first mention of the IIR on line 26:

" The IIR data along the lidar ground track provides additional information about cloud particle size using an improved split window technique (Garnier et al., 2012, 2013). "

Line 19: "HOI" has already been defined on Line 29 on Page 3.

> On line 19 we deleted the "hereafter to be called horizontally-oriented ice" since it is redundant to line 20 on page 3.

Lines 10-15: It would be helpful to the reader to discuss a range of expected differences between the molecular-only layer attenuated backscatter and the total layer attenuated backscatter, so that the assumption that $\delta_p(z) \cong \delta_v(z)$ is clear.

> Unfortunately, there is no simple way to characterize the range of the equivalent layer-integrated molecular backscatter, nor the difference between estimated particulate backscatter using the Hu equation (Eq. 5 in this paper) and the actual particulate backscatter. However, Hu et al. 2009 who developed the original threshold say this, "For optically thin clouds with extinction coefficients less than 0.2 km$^{-1}$, the molecular backscatter accounts for more than 10% of the 532-nm lidar return. The errors in the estimated particulate depolarization ratio can be as high as 0.1." Since this might be useful in thinking about the molecular contribution, this discussion has been added to the paper.

Line 37: remove closing parenthesis after Mioche et al., 2010.

> Done.

Line 22: Replace ", however this has been corrected in V4.5 and" with ". However, this has been corrected in V4.5, and"

> Done.

Line 23-24: Rewrite to "If the centroid temperature is > 0 $^{\circ}$C the layer will be assigned as low-confidence water, and occurs only rarely in < 0.1% of layers detected at 5-80 km."

> This suggestion is much better, thank you - done.

Page 14 Line 15 (and throughout): Use consistent terminology. Either "viewing angle" or "view". Some readers may confuse "view" for "field of view" taken out of context.

> Thank you, we can see how this is confusing.  After searching for the word "view" throughout the document, we have replaced "view" with "viewing angle" throughout the document where we mean viewing angle.  In a few instances we really do mean to say "view" and not "viewing angle" or "field of view".  These remain as "view".

Is the color scale for the CALIOP 532 nm backscatter curtain the standard one used in quicklooks (as in Figure 12a)? It appears that either the color scale is shifted or small values are being masked out, so please include a color bar or note about the values presented. There is no appearance of "total attenuation" underneath the clouds in Figure 11 as there is in Figure 12.

> This is a good catch. When William Straka blended the CALIOP and VIIRS data together to produce the image he eliminated the attenuated areas. Here is the amended Figure11 caption:

" Overview of "Superstorm" Sandy, with VIIRS day/night image and CALIPSO Sandy overpass on the left-hand side, and then on the right-hand side is a view from the west, with the corresponding CALIOP backscatter browse image superimposed. The CALIOP backscatter color scale used is the same as that used for standard browse images except that attenuated areas under the storm cloud tops are shown as background. This image was provided by William Straka III, at the University of Wisconsin."

Line 20: Fix the notation of "67deg N – 67deg S" to a symbolic representation of degrees (As in the caption for Figure 16).

> Done.